# Experimental protocol for sea level projections from ISMIP6 standalone ice sheet models

Sophie Nowicki[1], Heiko Goelzer[2,3], Helene Seroussi[4], Anthony J. Payne[5], William H. Lipscomb[6], Ayako Abe-Ouchi[7], Cecile Agosta[8], Patrick Alexander[9,10], Xylar S. Asay-Davis[11], Alice Barthel[11], Thomas J. Bracegirdle[12], Richard Cullather[1], Denis Felikson[1], Xavier Fettweis[13], Jonathan M. Gregory[14,15,16], Tore Hatterman[17,18], Nicolas C. Jourdain[19], Peter Kuipers Munneke[2], Eric Larour[4], Christopher M. Little[20], Mathieu Morlighem[21], Isabel Nias[1,22], Andrew Shepherd[23], Erika Simon[1], Donald Slater[24], Robin S. Smith[14,15], Fiammetta Straneo[24], Luke D. Trusel[25], Michiel R. van den Broeke[2] and Roderik van de Wal[2]

[1]NASA Goddard Space Flight Center, Greenbelt, MD, USA
[2]Institute for Marine and Atmospheric research Utrecht, Utrecht University, The Netherlands
[3]Laboratoire de Glaciologie, Université Libre de Bruxelles, Brussels, Belgium
[4]Jet Propulsion Laboratory, California Institute of Technology, Pasadena, CA, USA
[5]School of Geographical Sciences, University of Bristol, Bristol, UK
[6]Climate and Global Dynamics Laboratory, National Center for Atmospheric Research, Boulder, CO, USA
[7]Atmosphere and Ocean Research Institute, The University of Tokyo, Kashiwa-shi, Chiba, Japan
[8]Laboratoire des sciences du climat et de l'environnement, LSCE-IPSL, CEA-CNRS-UVSQ, Université Paris-Saclay, France
[9]Lamont-Doherty Earth Observatory, Columbia University, Palisades, NY, USA
[10]Nasa Goddard Institute for Space Studies, New York, NY, USA
[11]Los Alamos National Laboratory, NM, USA
[12]British Antarctic Survey, Cambridge, UK
[13]Laboratory of Climatology, Department of Geography, University of Liège, Liège, Belgium
[14]National Centre for Atmospheric Science, University of Reading, Reading, UK
[15]Department of Meteorology, University of Reading, Reading, UK.
[16]Met Office Hadley Centre, Exeter, UK
[17]Norwegian Polar Institute, Tromsø, Norway
[18]Energy and Climate Group, Department of Physics and Technology, The Arctic University – University of Tromsø, Norway
[19]Univ. Grenoble Alpes/CNRS/IRD/G-INP, Institut des Géosciences de l'Environnement, France
[20]Atmospheric and Environmental Research Inc., Lexington, Massachusetts, USA
[21]Department of Earth System Science, University of California Irvine, Irvine, CA, USA
[22]Department of Geography and Planning, School of Environmental Sciences, University of Liverpool, Liverpool, UK

[23]School of Earth and Environment, University of Leeds, Leeds, UK
[24]Scripps Institution of Oceanography, University of California, San Diego, La Jolla, CA, USA
[25]Department of Geography, Penn State University, University Park, PA, USA

*Correspondence to*: Sophie Nowicki (sophie.nowicki@nasa.gov)

## Abstract

Projection of the contribution of ice sheets to sea-level change as part of the Coupled Model Intercomparison Project – phase 6 (CMIP6) takes the form of simulations from coupled ice-sheet-climate models and standalone ice sheet models, overseen by the Ice Sheet Model Intercomparison Project for CMIP6 (ISMIP6). This paper describes the experimental setup for process-based sea-level change projections to be performed with standalone Greenland and Antarctic ice sheet models in the context of ISMIP6. The ISMIP6 protocol relies on a suite of polar atmospheric and oceanic CMIP-based forcing for ice sheet models, in order to explore the uncertainty in projected sea-level change due to future emissions scenarios, CMIP models, ice sheet models, and parameterizations for ice-ocean interactions. We describe here the approach taken for defining the suite of ISMIP6 standalone ice sheet simulations, document the experimental framework and implementation, as well as present an overview of the ISMIP6 forcing to be used by participating ice sheet modeling groups.

# 1    Introduction

The Ice Sheet Model Intercomparison Project for CMIP6 (ISMIP6) is a targeted activity of the Climate and Cryosphere (CliC) project of the World Climate Research Project (WCRP) and has been formally endorsed by the Coupled Model Intercomparison Project – Phase 6 (CMIP6, Eyring et al., 2016). Its aim is to provide process-based projections of the sea-level contributions from the Greenland and Antarctic ice sheets that are tightly linked to the wider suite of CMIP6 climate projections and employ forcing from the CMIP atmosphere-ocean general circulation model (AOGCM) ensemble. Nowicki et al. (2016) describe the overall design of ISMIP6, which includes an assessment of the impact of initial conditions on projections (ISMIP6-initMIP, Goelzer et al., 2018; Seroussi et al., 2019), experiments in which ice-sheet models are fully coupled within Earth-system models (ISM-ESMs), and experiments with standalone ice sheet models (ISM) forced by output from the CMIP AOGCM.

ISMs numerically simulate the dynamic flow of glacial ice – from the continental interior, across geographical distances, potentially transitioning to floating ice shelves, prior to terminating into the global ocean. An ISM receives input fields in the form of accumulating snowfall, surface and ocean temperature, and other time-varying conditions describing the ice sheet surface and its lateral boundaries, and it provides output fields of ice velocities and the distribution of ice mass. In a coupled ISM-ESM framework, the input fields become functions of how the ice sheets vary with time. Changes in the ice sheet topography can influence the atmospheric circulation, while the selective discharge of ice and meltwater may alter ocean circulation. The coupling of dynamical ice sheet models with ESMs is highly complex, as a mismatch between the relatively high spatial resolution and long integration time step of the ISM, and the relatively coarse spatial resolution and short integration time step of the ESM atmosphere and ocean fields must be negotiated (Vizcaino, 2014; Fyke et al., 2018). To date, only a limited number of ESM have been coupled with ISMs, which motivates the need for simulations with state-of-the-art standalone ISMs in order to explore the uncertainty in projected sea-level change.

Nowicki et al. (2016) described a number of possible avenues for the standalone component of ISMIP6 but was limited in its final protocol design because, at the time, the CMIP6 simulations had not started. Subsequent to the publication of Nowicki et al. (2016), ISMIP6 formed focus groups to evaluate the polar climate in the CMIP AOGCMs and to finalize how the output of CMIP AOGCMs would be translated into forcing for ice sheet models. Here we present a revised version of the ISMIP6 protocol, based on the improved understanding of ISM needs and CMIP6 AOGCM outputs. The complex issues of providing the offline AOGCM output at a high spatial resolution suitable for ISM modeling needed to be addressed in a uniform, standardized manner that would allow broad participation from the current generation of ice sheet models. Specific challenges included i) the translation of the various AOGCM resolutions and grids to the various ISM grid resolutions, ii) the poor representation of steep gradients in the surface topography of the ice sheet margins within AOGCMs, which underestimates large gradients in atmospheric forcing, iii) the quality of AOGCM polar climate and iv) the mismatch in the spatial extents of ice sheets within the AOGCMs and initialized ice sheet extent within certain ISMs. Additionally, oceanic variables from AOGCMs needed to be extrapolated from continental shelves to provide boundary conditions underneath ice shelves and at the calving front, as AOGCMs typically do not resolve ice shelf cavities and proglacial fjords. The ability to provide boundary conditions to ISM modeling groups in a timely manner is another factor that influenced the final ISMIP6 protocol design. The particular implementation of ice-atmosphere and ice-ocean interactions within each participating ISM had to be considered and these ISM specific constraints on the protocol were guided from lessons learned from the ISMIP6 initMIP efforts (Goelzer et al., 2018; Seroussi et al., 2019). Finally, the protocol needed to explore the uncertainty in sea-level projections due to the choice of emissions scenario, the choice of CMIP AOGCM, the ice sheet model physics (structural uncertainty) and how poorly constrained parameters within the model (parameter uncertainty), as well as uncertainty due to formulation of ice-ocean interactions.

This paper describes the detailed experimental protocol used for standalone experiments with ice-sheet models of Greenland and Antarctica using forcing from CMIP AOGCMs, and presents the novel atmospheric and oceanic forcing datasets prepared by ISMIP6. We begin by providing an overview of the projection framework and the purpose of the simulations in Sect. 2. We next present the protocol for initializing the projections, including schematic experiments needed to explore the uncertainty in ice sheet evolution due to initial state and historical simulations in Sect. 3. The atmospheric forcing and implementations are then described in Sect. 4, the oceanic forcing and implementation in Sect. 5, and Antarctic ice shelf fracture strategy in Sect. 6. We summarize the protocol and discuss the expected outcomes and impacts of ISMIP6 in Sect 7.

## 2 Overview of the projection setup and their purpose

Following the CMIP6 protocol (Eyring et al., 2016), all ISMIP6 projections start in January 2015 and end in December 2100. Although extensions beyond 2100 are available for some climate models in the CMIP5 archive and possible in the CMIP6 ScenarioMIP protocol (O'Neill et al., 2016), ISMIP6 focus remains on the end of the 21$^{st}$ century, as it is constrained by the availability and quality of the polar climate forcing. Beyond 2100, the ice sheets surface elevation will likely have deviated significantly from the fixed ice sheet elevation configuration used by the CMIP models, which may affect projected polar climate.

The projection setup strategy for the ISMIP6 standalone ISMs is illustrated in Fig. 1. For a given CMIP AOGCM and future climate scenario, an ISM atmospheric forcing dataset ultimately takes the form of surface mass balance (SMB) and surface temperature, while oceanic forcing data includes oceanic temperature and salinity in order to infer Antarctic ice shelf basal melt, and Greenland calving and frontal melt. For the Antarctic ice shelves, atmospheric properties may result in surface melting, which in turn can trigger ice shelf collapse (Trusel et al., 2015). An issue faced by ISMIP6 is the mismatch in spatial resolution and spatial extent between available AOGCM fields and ISM needs: AOGCMs do not generally resolve oceanic flow within the Greenland fjords or beneath the Antarctic ice shelves, and SMB varies rapidly over the steep topography at the ice sheet margins, but these SMB gradients are not captured by AOGCMs. The implication is that extrapolation and downscaling of atmospheric and oceanic AOGCM fields may be required to produce realistic ice sheet projections. For the Greenland ice sheet atmospheric fields, downscaling was done via the use of a regional climate model. For both Greenland and Antarctica, far field ocean temperature and salinity were extrapolated through Greenland fjords and beneath Antarctic ice shelves using rules that account for the blocking effects of bathymetric sills.

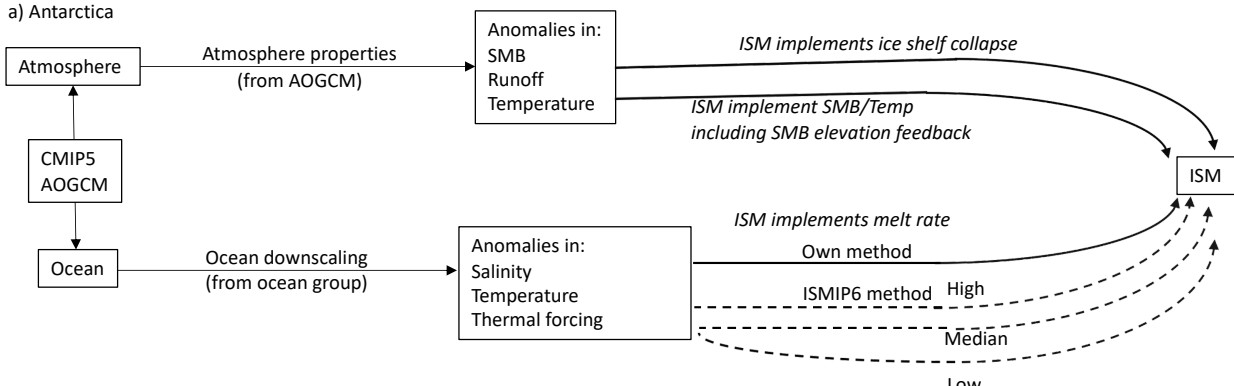

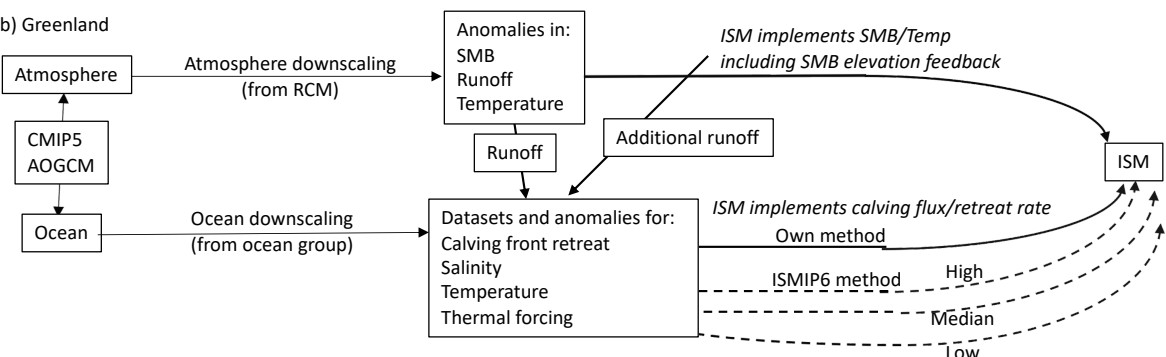

**Figure 1:** Overview of ISMIP6 a) Antarctic and b) Greenland projections framework, illustrating the strategy for translating the CMIP atmospheric and oceanic properties into climatic forcing for ice sheet models.

Ice-ocean interactions remains an active area of research, and the current generation of ice sheet models uses a variety of representations. For example, the Antarctic ice-ocean interaction representations range from simple linear relationships between oceanic temperature change and melt rate (eg. Rignot and Jacobs, 2002) to more complex parameterizations (see Favier et al. (2019) for a review). Similarly, the Greenland ice-ocean interactions in large scale ice sheet models range from ad-hoc methods (e.g. Price et al., 2011; Bindschadler et al., 2013; Goelzer et al., 2013; Nick et al., 2013; Furst et al., 2015) to estimated submarine melt rates (Aschwanden et al., 2019). The ISMIP6 protocol allows for sampling the uncertainty in the representation of ice-ocean interaction via the use of "open" and "standard" experiments (Sect. 5). In the "standard" experiments, ice sheet models implement parameterizations and forcings designed by the ocean focus groups. The standard experiments further test the parameter uncertainty in the basal melt formulations by sampling "low", "mid", and "high" values of the parameters in the melt-rate parameterizations for both ice sheets. In addition, for the Antarctic ice sheets, two calibrations are investigated (see Jourdain et al., under review): one based on observed mean sub-shelf basal melt over Antarctica (MeanAnt) and one based sub-shelf basal melt near the grounding line of Pine Island glacier (Pine Island Grounding Line calibration, PIGL). In the "open" experimental setup, ice sheet models can use their existing parameterization for ice-ocean interaction, driven by the provided extrapolated AOGCM oceanic forcing datasets. The open experiments further allow for inclusion of additional physical processes that are not taken into account in the standard framework. An example is exploring the Greenland ice sheet response to surface meltwater reaching the base of the ice sheet and affecting basal sliding. As there is disagreement on the implication of this process for ice sheet evolution (Shannon et al., 2013; Koziol and Arnold, 2018), this forcing is not part of the standard experiments, but could be explored in the open approach.

Future climate scenarios are defined in CMIP5 as Representative Concentration Pathways (RCPs, van Vuuren et al., 2011) and in CMIP6 as Shared Socioeconomic Pathways (SSPs, Eyring et al., 2016; O'Neill et al., 2016). The ISMIP6 protocol

samples the potential sea-level from ice sheets under two different climate scenarios: the high emission CMIP5 RCP8.5 and CMIP6 SSP5-8.5 scenarios are our primary focus, but the lower emission scenarios CMIP5 RCP2.6 and CMIP6 SSP1-26 are included for a few CMIP models. The focus on CMIP5 scenarios in this revised ISMIP6 protocol is due to the delay in CMIP6 model simulations, which prevents a full analysis of the CMIP6 models. The use of the CMIP multi-model mean for forcing

ISMs is not feasible in our experimental protocol, as it is not be possible to downscale atmospheric fields via regional climate models (RCMs) with a multi-model mean. It is also not feasible to use all of the CMIP models, due to both the time and computational effort needed to prepare the forcing dataset, as well as the time and computational effort required for running the ISMs. The ISMIP6 strategy is to sample the CMIP ensemble in order to select a manageable number of CMIP models for our projections that is representative of the spread in the full CMIP ensemble. As described in Barthel et al. (2020), six CMIP5

AOGCMs were selected per ice sheets based on the following criteria: i) present-day polar climate in agreement with observations (evaluated by model biases over the historical period, for example Agosta et al. (2015)), ii) sampling a diversity of future climate (evaluated by difference in projections and code similarities), and iii) a focus on models with RCP8.5 and RCP2.6 which also have the fields required for RCM downscaling. The CMIP5 models were selected independently for the Greenland and Antarctic ice sheets, using atmospheric and oceanic metrics appropriate for each ice sheets. Four of the CMIP5

models were chosen to be the same for both ice sheets (CSIRO-MK3.6.0, HadGEM2-ES, IPSL-CM5A-MR, NorESM1-M), the fifth choices were closely related (MIROC5 for Greenland and MIROC-ESM-CHEM for Antarctica), and the sixth choices were unrelated (ACCESS1.3 for Greenland and CCSM4 for Antarctica). For more information, see Table 1. The key characteristics of models taking part in CMIP5 are summarized in Tables 9.1 and 9.A.1 of Flato et al. (2013). As CMIP6 models started to become available in late spring/summer 2019, ISMIP6 selected four CMIP6 AOGCMs (CESM2, CNRM-

CM6-1, CNRM-ESM2-1, UKESM1-0-LL) based solely on their availability and the fact that two of these models would be taking part in the coupled climate-ice sheet component of ISMIP6, which allows for future scientific analysis of the difference in projection from standalone ISMs versus ISMs that are fully coupled to climate models.

**Table 1.** CMIP Model used to obtain atmospheric and oceanic forcing for the Greenland and Antarctic ice sheets

| Model Name | Institution | Main Reference(s) | Use in ISMI6 |
|---|---|---|---|
| CMIP5 Models | | | |
| ACCESS1.3 | Commonwealth Scientific and Industrial Research Organization and Bureau of Meteorology, Australia | Bi et al., 2013; Dix et al., 2013 | Greenland |
| CCSM4 | US National Centre for Atmospheric Research | Gent et al., 2011 | Antarctica |
| CSIRO-Mk3.6.0 | Queensland Climate Change Centre of Excellence and Commonwealth Scientific and Industrial Research Organisation | Collier et al., 2011 Rotstayn et al., 2012 | Antarctica, Greenland |
| HadGEM2-ES | UK Met Office Hadley Centre | Collins et al., 2011; Martin et al., 2011 | Antarctica, Greenland |
| IPSL-CM5A-MR | Institut Pierre Simon Laplace | Dufresne et al., 2012 | Antarctica, Greenland |
| MIROC5 | University of Tokyo, National Institute for Environmental Studies and Japan Agency for Marine-Earth Science and Technology | Watanabe et al., 2010 | Greenland |
| MIROC-ESM-CHEM | University of Tokyo, National Institute for Environmental Studies and Japan Agency for Marine-Earth Science and Technology | Watanabe et al., 2011 | Antarctica |

| NorESM1-M | Norwegian Climate Centre | Iversen et al., 2013 | Greenland, Antarctica |
|---|---|---|---|
| | CMIP6 Models | | |
| CESM2 | US National Centre for Atmospheric Research | Danabasoglu et al., 2020 | Greenland, Antarctica |
| CNRM-CM6-1 | Centre National de Recherche Meteorologiques and Cerfacs | Voldoire et al., 2019 | Greenland, Antarctica |
| CNRM-ESM2-1 | Centre National de Recherche Meteorologiques and Cerfacs | Séférian et al., 2019 | Greenland, Antarctica |
| UKESM1-0-LL | UK Met Office and Natural Environment Research Council | Sellar et al., 2019 | Greenland, Antarctica |

The projection protocol consists of "core", or Tier 1, experiments, which modeling groups are required to perform, and "targeted experiments". The targeted experiments are optional, and further divided into higher priority Tier 2 and lower priority Tier 3 experiments. Core experiments are designed to explore the range of CMIP5 model uncertainty with three AOGCMs under two future emissions scenarios, the impact of Antarctic ice shelf fracture, and the uncertainty in ocean parameters for groups that participate in the standard oceanic protocol. Groups that can participate in both the open and standard implementations of ocean forcing are encouraged to do so. This set up results in five open and eight standard projections for Antarctica (Table 2); and four open and six standard experiments for Greenland (Table 3). The Tier 2 experiments consist of the remainder of the three CMIP5 AOGCMs and the three CMIP6 models. Tier 3 experiments further explore the uncertainty in the standard ocean parameterization, the impact of ice shelf fracturing, and simulations driven by atmosphere only forcing (no change in ocean) and ocean only forcing (no change in atmosphere) to help understand the source of mass loss from the corresponding full simulations, in which changes are due to both atmosphere and ocean. The complete list of Tier 2 and Tier 3 experiments is presented in Appendix A. This mix of core and targeted experiments follows the approach taken in Shannon et al. (2013): it provides a flexible framework that allows less computationally expensive models to explore the full set of experiments, allowing modelers to choose to focus on a certain aspects of the protocol that fits their research interests, while ensuring that all groups perform a subset of identical experiments.

Participating groups may decide to investigate the impact of ice sheet model uncertainty in projections via different model set up choices. These include exploring mesh resolution as well as model parameterizations such as the basal sliding law, parameters in ice sheet flow approximation, and ice shelf basal melt parameterization. Unlike the original initMIP effort, SMB and bedrock adjustment in response to a changing ice sheet is allowed. In some cases, it may be necessary to treat these modeling set up choices as different models, and repeat the initialization method, the initMIP experiments, the historical and control runs described below.

**Table 2**. Antarctic core (Tier 1) experiments.

| Experiment ID | RCP | AOGCM | Standard/ Open | Ocean Forcing | Fracture | Note |
|---|---|---|---|---|---|---|
| exp01 | 8.5 | NorESM1-M | Open | Medium | None | Low atmospheric change and mid-to-high ocean warming |
| exp02 | 8.5 | MIROC-ESM-CHEM | Open | Medium | None | High atmospheric changes and median ocean warming |
| exp03 | 2.6 | NorESM1-M | Open | Medium | None | Low atmospheric change and mid-to-high ocean warming |

| exp04 | 8.5 | CCSM4 | Open | Medium | None | Large atmospheric warming and variable regional ocean warming |
| exp05 | 8.5 | NorESM1-M | Standard | MeanAnt Medium | None | Low atmospheric change and mid-to-high ocean warming |
| exp06 | 8.5 | MIROC-ESM-CHEM | Standard | MeanAnt Medium | None | High atmospheric changes and median ocean warming |
| exp07 | 2.6 | NorESM1-M | Standard | MeanAnt Medium | None | Low atmospheric change and mid-to-high ocean warming |
| exp08 | 8.5 | CCSM4 | Standard | MeanAnt Medium | None | Large atmospheric warming and variable regional ocean warming |
| exp09 | 8.5 | NorESM1-M | Standard | MeanAnt High | None | Ocean forcing uncertainty, using 95th percentile values |
| exp10 | 8.5 | NorESM1-M | Standard | MeanAnt Low | None | Ocean forcing uncertainty, using 5th percentile values |
| exp11 | 8.5 | CCSM4 | Open | Medium | Yes | Experiment with ice shelf hydrofracture |
| exp12 | 8.5 | CCSM4 | Standard | MeanAnt Medium | Yes | Experiment with ice shelf hydrofracture |
| exp13 | 8.5 | NorESM1-M | Standard | PIGL Medium | None | Ocean forcing uncertainty, using PIGL gamma calibration |

**Table 3.** Greenland core (Tier 1) experiments.

| Experiment ID | RCP | AOGCM | Standard/ Open | Ocean Forcing | Note |
|---|---|---|---|---|---|
| exp01 | 8.5 | MIROC5 | Open | Medium | Expected largest response to SMB, median ocean warming |
| exp02 | 8.5 | NorESM1-M | Open | Medium | Low atmosphere change, low ocean warming |
| exp03 | 2.6 | MIROC5 | Open | Medium | Expected largest response to SMB, median ocean warming |
| exp04 | 8.5 | HadGEM2-ES | Open | Medium | Expected median response to SMB, median ocean warming |
| exp05 | 8.5 | MIROC5 | Standard | Medium | Expected largest response to SMB, median ocean warming |
| exp06 | 8.5 | NorESM1-M | Standard | Medium | Low atmosphere changes, low ocean warming |
| exp07 | 2.6 | MIROC5 | Standard | Medium | Expected largest response to SMB, median ocean warming |
| exp08 | 8.5 | HadGEM2-ES | Standard | Medium | Expected median response to SMB, median ocean warming |
| exp09 | 8.5 | MIROC5 | Standard | High | Ocean forcing uncertainty |
| exp10 | 8.5 | MIROC5 | Standard | Low | Ocean forcing uncertainty |

Ice sheet modeling groups are requested to submit the variables listed in Appendix B, as long as these variables are applicable to their models. These consist of state variables (such as ice thickness), flux variables (such as SMB), as well as integrated scalar values (such as total ice sheet wide SMB flux). To facilitate the analysis and intercomparison, groups should save their model output on one of the ISMIP6 grids with spatial resolution closest to the model native grid (see Appendix B for details).

## 3    Initial state, control runs and historical run

Ice sheet model initial states are typically obtained via different methods: long interglacial spinup or data assimilation of present-day observations, as well as hybrid combinations of these two methods (Nowicki et al., 2013a.b; Goelzer et al., 2017; Pattyn et al., 2017). Interglacial spinups have the advantage of obtaining an ice sheet that can capture transients due to past climatic conditions, but the disadvantage of producing an ice sheet geometry that may differ from present day (Seroussi et al., 2013). Assimilation methods, on the other end, capture the present-day geometry but projections often suffer from unrealistic drifts due to the model responding to inconsistencies in the input datasets. Time-dependent data assimilation methods allow

for more realistic transients, but to date have been limited to regional studies or synthetic ice sheet setup (e.g. Goldberg et al., 2015; Gillet-Chaulet, 2020). Other methods include combinations of these techniques, as demonstrated in the initMIP Greenland and Antarctica efforts (Goelzer et al., 2018; Seroussi et al., 2019). These multiple approaches for initialization, and the use of different observations for assimilation methods, creates a challenge in the design protocol, as the initial state date becomes model specific, and ranged from the 1990s to the 2010s in the initMIP efforts, for example.

Modeling groups taking part in the ISMIP6 projections are free to reuse their "initial state" submitted as part of initMIP or create a new initial state. In the latter case, groups are asked to re-run the 100 years long initMIP schematic experiments: anomalies in surface mass balance ("asmb") for Greenland and Antarctica, and anomalies in ice shelf basal melt ("bsmb") for Antarctica; as well as a control run ("ctrl"). This control run is needed to capture any drift present in the projections as a result of the initialization method. The control run is implemented as a forward run without any anomaly forcing, such that for example any surface mass balance used in the initialization would continue unchanged. The control run begins from the same initial state as the initMIP schematic experiments and the historical run, and lasts as long as the initMIP experiments and the projections. The control run may need to be extended from an original initMIP submission (where the control was set to 100 yrs) in order to cover the time period from the initial state to the end of the projections. Table 4 illustrates typical set up, and time span for the ISMIP6 protocol.

The "historical run" bridges the gap between the time of the ice sheet "initial state" and the "projections start state" in January 2015. The projections then branch from a single historical run for each ice sheet model. Because the time of each ice sheet model's initial state varies, the duration of the historical run will therefore also vary between models. Ice sheet modelling groups are left to decide how to perform the historical run and bring their models to the "projection start state", a choice motivated by i) the distinct initialization procedures used in the ice sheet modelling community, ii) the lack of known set of historical atmospheric and oceanic forcing that can reproduce observed changes, due in part to the limited observational record, and iii) the challenges associated with our revised strategy of using multiple CMIP models and scenarios to sample the uncertainty in future climate. The latter would require multiple historical runs from each ice sheet model, which may then result in distinct projection start state for a given ice sheet model, complicating the projection forcing strategy as well as interpretation of the simulations. Nonetheless, AOGCM derived historical datasets are provided for each AOGCM projection dataset. Modeling groups are free to use one of the AOGCM historical dataset, or a reanalysis, or a combination of multiple dataset. To test the impact of the choice of historical dataset on the projections and associated model drift, groups are required to submit a "projection control". This simulation is also an unforced ice sheet model run (implemented with zero anomalies), which starts from the ice sheet "projection start state" (January 2015) and runs until December 2100.

**Table 4.** Initialization experiments and examples of duration of experiment for different choices for "initial state" and "projection start state". Modeling groups are free to choose "initial state" dates that are not indicated in this Table, but the "projection start state" should always correspond to January 2015. *denotes experiments that are only needed if the initial state used for the projections is different than that submitted for the initMIP effort. The ctrl may need to be extended from an original initMIP submission.

| Experiment ID | Note | Start 1 (duration) | Start 2 (duration) | Start 3 (duration) |
|---|---|---|---|---|
| N/A | Initial state date (result of initialization) | 01/01/1980 | 01/01/2005 | 01/01/2005 |
| ctrl* | Unforced control run starting from initial state, needed for model drift evaluation due to initialization | 01/01/1980 (120 years) | 01/01/2005 (100 years) | 01/01/2015 (100 years) |
| historical | Historical simulation, needed to bring model from initial state to projection start date | 01/01/1980 (35 years) | 01/01/2005 (10 years) | N/A (0 years) |
| ctrl_proj | Unforced control run starting from the projection start date, needed for model drift evaluation due to historical | 01/01/2015 (86 years) | 01/01/2015 (86 years) | 01/01/2015 (86 years) |
| asmb* | initMIP prescribed surface mass balance simulation (Antarctica and Greenland) | 01/01/1980 (100 years) | 01/01/2005 (100 years) | 01/01/2015 (100 years) |

| | | | | | |
|---|---|---|---|---|---|
| bsmb* | initMIP prescribed basal mass balance simulation (Antarctica only) | 01/01/1980 (100 years) | 01/01/2005 (100 years) | 01/01/2015 (100 years) |

## 4    Atmospheric forcing and implementation

Atmospheric forcing for standalone ice sheet model simulations consist of surface mass balance (SMB) and surface temperature derived from CMIP AOGCMs. SMB provides mass gain from accumulation (snow and rain) and mass loss from ablation or surface melting to the ice sheet model. Current AOGCM outputs can be directly used to compute SMB, but this approach does not capture the narrow peripheral region with steep SMB gradients, which are key for ice sheet simulations.  In this case, AOGCM climate can be downscaled using high resolution regional climate models (RCMs), which is the technique used for ISMIP6 Greenland projections.  RCMs currently provide more realistic surface climate than the direct output of CMIP5 AOGCMs for both ice sheets (Fettweis et al., 2013; Noel et al., 2018; van Wessem et al., 2018; Agosta et al., 2019). In the future, it may be possible to bypass the use of RCMs for downscaling, as a new generation of climate models have implemented multiple elevation classes (CESM, Lipscomb et al., 2013; UKESM1, Sellar et al., 2019; ModelE, Fischer et al., 2014), allowing SMB to be computed at multiple elevations within an horizontal grid cell in order to capture SMB gradients, as well as improvements in the parameterization of polar surface processes in AOGCMs (e.g. Cullather et al., 2014; van Kampenhout et al., 2017; Alexander et al., 2019).  The use of CMIP5 models for this protocol therefore requires AOGCM to be selected for their skills in simulating forcing fields for RCMs for both Greenland and Antarctica, instead of their skills in simulating SMB.  Due to time constraints and computational demands with the use of RCMs, the atmospheric forcing for the ice sheet simulations were only obtained with an RCM for the Greenland ice sheet, and derived directly from the AOGCM output for the Antarctic ice sheet simulations.

Regardless of the methodology chosen to obtain surface forcing for the ice sheet models from CMIP AOGCM, the strategy of investigating the uncertainty due to CMIP climate models and scenarios (which requires the use of multiple  CMIP AOGCMs), as well as the distinct initialization methods used by ice sheet models (which uses diverse SMB and temperature sources), prevents the direct application of SMB and surface temperature as a boundary condition for ice sheet models. Instead, surface forcings are implemented via annual SMB and temperature anomalies. This choice assumes that the inter-annual variability in SMB and temperature anomalies is greater than any differences in SMB and temperature climatologies from both the CMIP AOGCMs and the climatologies used in the initialization of ice sheet models.

### 4.1.   Antarctic atmospheric forcing and implementation

For the Antarctic ice sheet, ISMIP6 provides yearly averaged anomalies in SMB (computed as precipitation minus evaporation minus runoff) and surface temperature, along with the respective climatologies for SMB and temperature used to compute the anomalies. The data were prepared by ISMIP6 using direct input from the CMIP models listed in Table 1 and interpolated onto the ISMIP6 input grids. More information on how the datasets were prepared is available in Appendix C. The anomalies were computed as:

$$aSMB_{AOGCM}(x,y,t) = SMB_{AOGCM}(x,y,t) - SMB_{CLIM,AOGCM}(x,y) \tag{1}$$

and

$$aT_{AOGCM}(x,y,t) = T_{AOGCM}(x,y,t) - T_{CLIM,AOGCM}(x,y) \tag{2}$$

where $aSMB_{AOGCM}(x,y,t)$ and $aT_{AOGCM}(x,y,t)$ are the anomalies in SMB and temperature, $SMB_{AOGCM}(x,y,t)$ and $T_{AOGCM}(x,y,t)$ are the annual SMB and temperature for a given AOGCM, while $SMB_{CLIM,AOGCM}(x,y)$ and $T_{CLIM,AOGCM}(x,y)$ the climatologies. The climatogolies were computed by taking the mean values of $SMB_{AOGCM}$ and $T_{AOGCM}$ over the Antarctic reference period (January 1995 to December 2014). The anomaly datasets cover the time period of 1950 to 2100.

During the simulations, ice sheet models need to reintroduce the climatology that best fit their simulation, and compute surface input ($SMB_{ISM,AOGCM}(x, y, t)$ and $T_{ISM,AOGCM}(x, y, t)$) as:

$$SMB_{ISM,AOGCM}(x, y, t) = SMB_{REF}(x, y) + aSMB_{AOGCM}(x, y, t) \qquad (3)$$

and

$$T_{ISM,AOGCM}(x, y, t) = T_{REF}(x, y) + aT_{AOGCM}(x, y, t) \qquad (4)$$

where $SMB_{REF}(x, y)$ and $T_{REF}(x, y)$ are the SMB and temperature that the ice sheet model would have used over the reference period, which is the same for all core and targeted experiments. $SMB_{REF}(x, y)$ and $T_{REF}(x, y)$ should be computed as the average over the reference period for time-dependent SMB and temperatures, or simply set to the climatology used in the case of time-independent input. Note that the anomalies are constant over the entire year and changes step wise at the beginning of every year.

The surface temperatures and SMB fields to be used in forcing the ISM simulations are illustrated in Figs. 2 and 3 respectively. Time series of values averaged over the fixed Antarctic ice sheet mask under RCP2.6, RCP8.5, SSP1-2.6, and SSP5-8.5 scenarios for all Tier experiments are shown, while the spatial patterns of mean anomalies from 2081-2100 are shown for Tier 1 experiments only (Table 2). These datasets were generated by ISMIP6 and it is their first presentation in the literature. We note that CCSM4 has a finer native model resolution (0.94° × 1.25° grid, see Table C1 in Appendix C) compared to NorESM1-M (1.91° × 2.50°) and MIROC-ESM-CHEM (2.81° × 2.81°). As expected, the SMB and temperature anomalies are correlated: CMIP models that project warmer surface conditions (Fig. 2a), also project an increased SMB (Fig. 3a). From the Tier 1 simulations, the largest temperature increases are found in CCSM4 and MIROC-ESM-CHEM fields under the RCP8.5 scenario. The temperature anomalies from these models are about 3.6 to 6.4 K warmer in the 2081-2100 period than at the start of the projection. This contrasts with the temperature fields of the selected NorESM1-M CMIP5 model, which indicates negligible temperature increases by the end of the century for the RCP2.6 scenario, and a 2K to 3.8K increase for the RCP8.5 scenario. As with the continent-averaged time series, the spatial patterns of temperature anomalies for the NorESM1-M RCP2.6 output for the 2081 to 2100 period are similarly muted (Fig. 2b), with greatest warming over the Peninsula, Ronne and Amery ice shelves, and over the East Antarctic plateau under RCP8.5 (Fig. 2c). The MIROC-ESM-CHEM and CCSM4 models (Fig. 2d,e) under the RCP8.5 scenario project similar spatial patterns of temperature anomalies as with the NorESM1-M, but with greater magnitude. The MIROC-ESM-CHEM output indicates greater warming than for the CCSM4 over the Peninsula, the ice shelves adjacent to Dronning Maud Land, and Enderby and Kemp Land glaciers in East Antarctica, and over the Getz ice shelf and the ice shelves fed by Marie Byrd Land glaciers in West Antarctica. CCSM4 shows less warming over the steep margins of the ice sheet.

The coarser grid of MIROC-ESM-CHEM impacts the spatial distribution of projected SMB anomalies (Fig. 3d), with negative values over Marie Byrd Land, the Getz and Abbot ice shelves, and glaciers feeding the Ronne and Fichner ice shelves in East Antarctica, as well as the Peninsula, and in East Antarctica over the Brunt ice shelf, Endery Land, and on both flanks of the Amery ice shelf, due to large surface runoff in these regions. The projected SMB anomalies in MIROC-ESM-CHEM are negligible over the East Antarctic plateau and increase towards the coast. This pattern of minimal SMB change over the interior plateau and increased SMB for coastal regions is also captured in CCSM4 and the NorESM1-M models under RCP8.5 (Fig. 3c,e), with the particular exception of the northern Antarctic Peninsula, where the anomalies are negative due to sufficiently warm conditions that promote ablation. The relatively fine grid used by CCSM4 allows for more spatial variations in SMB anomalies, particularly over the steep ice sheet margins. Aside from negative SMB anomalies over the tip of the Peninsula, the NorESM1-M model indicates little change with time in conditions under RCP2.6 (Fig. 3b), which is also reflected in the mean ice sheet-wide SMB anomalies (Fig. 3a). The large areas with negative SMB anomalies projected by the MIROC-ESM-CHEM simulation balance with increases in other regions such that the ice sheet-averaged SMB anomalies are comparable to those of the NorESM1-M model under RCP8.5. The CCSM4 is the Tier 1 model with largest ice sheet-wide projected increase in SMB, which is comparable to the Tier 2 CMIP6 models under SSP5-8.5 (Fig. 3a).

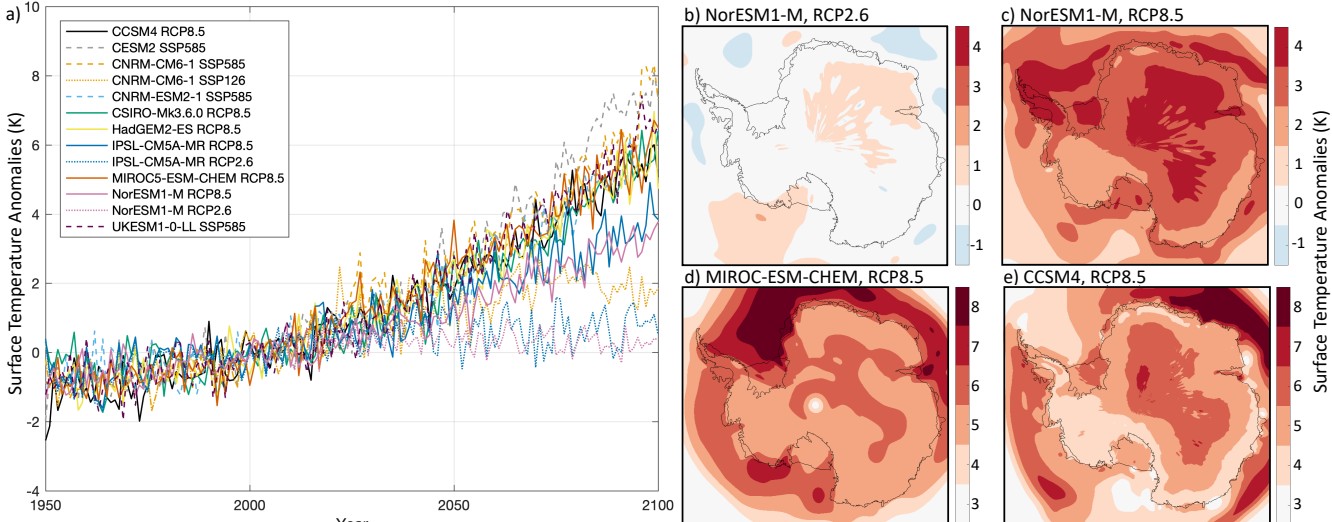

**Figure 2:** Surface temperature anomalies (K) over the Antarctic ice sheet under the RCP2.6, RCP8.5, SSP1-2.6 and SSP5-8.5 scenarios. a) Time series of mean surface temperature anomaly from 1950 to 2100 for all AOGCMs and scenarios selected, and (b-e) surface temperature anomaly over the time period 2081-2100 for b) NorESM1-M under RCP2.6, c) NorESM1-M under RCP8.5, d) MIROC-ESM-CHEM under RCP8.5, and e) CCSM4 under RCP8.5.

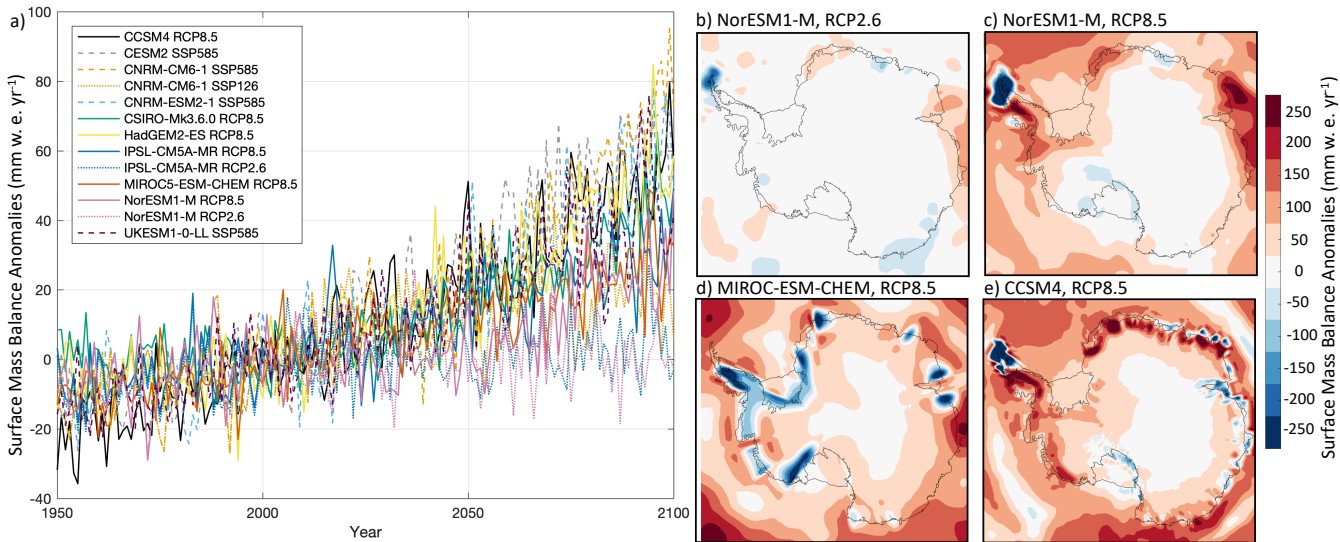

**Figure 3:** Surface mass balance anomalies (mm w. e. yr$^{-1}$) over the Antarctic ice sheet under the RCP2.6, RCP8.5, SSP1-2.6 and SSP5-8.5 scenarios. a) Time series of mean surface mass balance anomaly from 1950 to 2100 for all AOGCMs and scenarios selected, and (b-e) mean surface mass balance anomaly over the time period 2081-2100 for b) NorESM1-M under RCP2.6, c) NorESM1-M under RCP8.5, d) MIROC-ESM-CHEM under RCP8.5, and e) CCSM4 under RCP8.5.

### 4.2 Greenland atmospheric forcing and implementation

For the Greenland ice sheet, ISMIP6 generated surface forcing from CMIP AOGCMs that have been re-interpreted through the Modèle Atmosphérique Régionale (MAR) regional climate model (version 3.9.6, Delhasse et al., 2019). Although MAR uses a fixed topography for the Greenland ice sheet, it also allows for SMB-height feedback to be included in the ice sheet model simulations by providing vertical SMB and temperature gradients on each horizontal grid cell (Franco et al., 2012). The vertical gradients are also used to downscale the original MAR results computed on a 15 km grid to a finer 1 km grid, allowing for the resolution of steep topography. In addition, MAR calculates potential SMB and temperature in regions that are outside

the MAR ice sheet mask, allowing for surface forcing to be computed for ice sheet spatial extent that differs slightly from the MAR ice sheet mask.

For each MAR downscaled CMIP model, ISMIP6 provides annual anomalies in SMB and surface temperature from 1950 to 2100, the vertical SMB and temperature gradients over the same time period, as well as the respective climatologies computed over the reference period of January 1960 to December 1989. The reference period is distinct from the one for Antarctica, because the Greenland ice sheet is considered to have been in steady state with the climate during this period (e.g. Fettweis et al., 2017; Mouginot et al., 2019). The surface forcing anomalies were computed in a manner similar to Antarctica (Eq 1, 2), and need to be added to the ice sheet model climatology during the simulation. The implementation of surface forcing during the ice sheet simulation differs slightly from the set up for Antarctica, as the MAR computed vertical SMB and temperature gradients fields allows for implementation of SMB and temperature feedback in the Greenland framework. During the ice sheet projection, surface forcings are implemented as:

$$SMB_{ISM,RCM}(x,y,t) = SMB_{REF}(x,y) + aSMB_{RCM}(x,y,t) + \frac{dSMB_{RCM}(x,y,t)}{dz}\left(h_{ISM}(x,y,t) - h_{REF}(x,y)\right) \quad (5)$$

and

$$T_{ISM,RCM}(x,y,t) = T_{REF}(x,y) + aT_{RCM}(x,y,t) + \frac{dT_{RCM}(x,y,t)}{dz}\left(h_{ISM}(x,y,t) - h_{REF}(x,y)\right) \quad (6)$$

where $h_{ISM}(x,y,t)$ is the ice sheet model time-dependent surface elevation, $h_{REF}(x,y)$ is the ice sheet model surface elevation at the start of the projection, $\frac{dSMB_{RCM}(x,y,t)}{dz}$ the time-dependent SMB vertical gradient and $\frac{dT_{RCM}(x,y,t)}{dz}$ the time-dependent temperature vertical gradient.

The MAR SMB forcing can only be directly applied when the ice sheet model projection start state is close to the present-day geometry used in the MAR simulations. However, the initMIP-Greenland experiments (Goelzer et al., 2018) show that for some ice sheet models the present-day ice sheet can differ substantially from the observed ice sheet configuration, especially for models that initialize their ice sheet to present day via inter-glacial spinup. This can also be the case for models that use assimilation techniques and a long relaxation scheme that results in large geometric changes. In these cases, the surface forcing anomalies and vertical gradients should be corrected and remapped to the modeled ice sheet, using the technique described in Goelzer et al. (in review). The method uses the strong dependence of SMB and temperature on elevation, to remap the MAR field and reduce unphysical biases while preserving the overall surface forcing patterns. Once the surface forcings have been remapped by ISMIP6 to an individual ice sheet configuration, modelers should implement surface forcings in the same manner as described previously.

The MAR-derived surface temperature and SMB fields to be used in forcing the ISMs are shown in Figs. 4 and 5, respectively. Time series of ice sheet area-averaged values under the RCP2.6, RCP8.5, SSP1-2.6 and SSP5-8.5 scenarios for all Tier experiments over the 21st century were computed using the fixed Greenland ice sheet present-day area mask. The spatial patterns of anomalies averaged for the 2081-2100 period are illustrated using the Tier 1 models (Table 3). These datasets are specific to the ISMIP6 project and shown here for the first time in the literature. Although the time series of both surface temperature and SMB anomalies show considerable interannual variability, the experiments project an overall warming, and an associated negative SMB trend over the margins due to increased surface runoff. At the top of the range of Tier 1 models under the RCP8.5 scenario, the HadGEM2-ES projects an increase in surface temperature of about 7 K by 2100 (Fig. 4a), with greatest warming over the northeastern Greenland ice sheet and smaller values concentrated over the southern ice sheet margins that are characterized by steep topography (Fig. 4e). The patterns of mean SMB change for HadGEM2-ES (Fig. 5e) project increases over the interior of the ice sheet due to enhanced precipitation, and large negative SMB anomaly values over the periphery on the ice sheet. This general pattern of SMB change is also found with the MIROC5 and NorESM1-M models under RCP8.5 (Fig. 5c and 5e), but unlike the HadGEM2-ES, positive SMB anomalies are also located over the south plateau. The region is projected to warm less in the MIROC5 and NorESM1-M projections as compared to the HadGEM2-ES (Fig 4 c-e). For the entire Greenland ice sheet, the NorESM1-M project the lowest area-averaged increase in

surface temperature – and the smallest decrease in SMB – as compared to the MIROC5 and HadGEM2-ES integrations under RCP8.5 (Figs. 4a and 5a). This is also reflected in the spatial patterns of surface temperature (Fig. 4d), in which the NorESM1-M output indicates smaller temperature increases over the periphery of the ice sheet, but temperature increases comparable to the MIROC5 over the central western and southern regions.  For the selected simulations of the RCP2.6 emission scenario, surface temperatures are generally projected as slightly increased as compared to present-day conditions (Fig. 4b) but with regional variations: for the MIROC5 output, the northern Greenland ice sheet is projected to warm more than the southern region. The output indicates an increase in SMB for the interior of the ice sheet and decrease around the periphery. Areas of negative SMB anomalies are found to extend further inland as compared to the RCP8.5 integrations, despite the smaller magnitude (Fig. 5b,c).

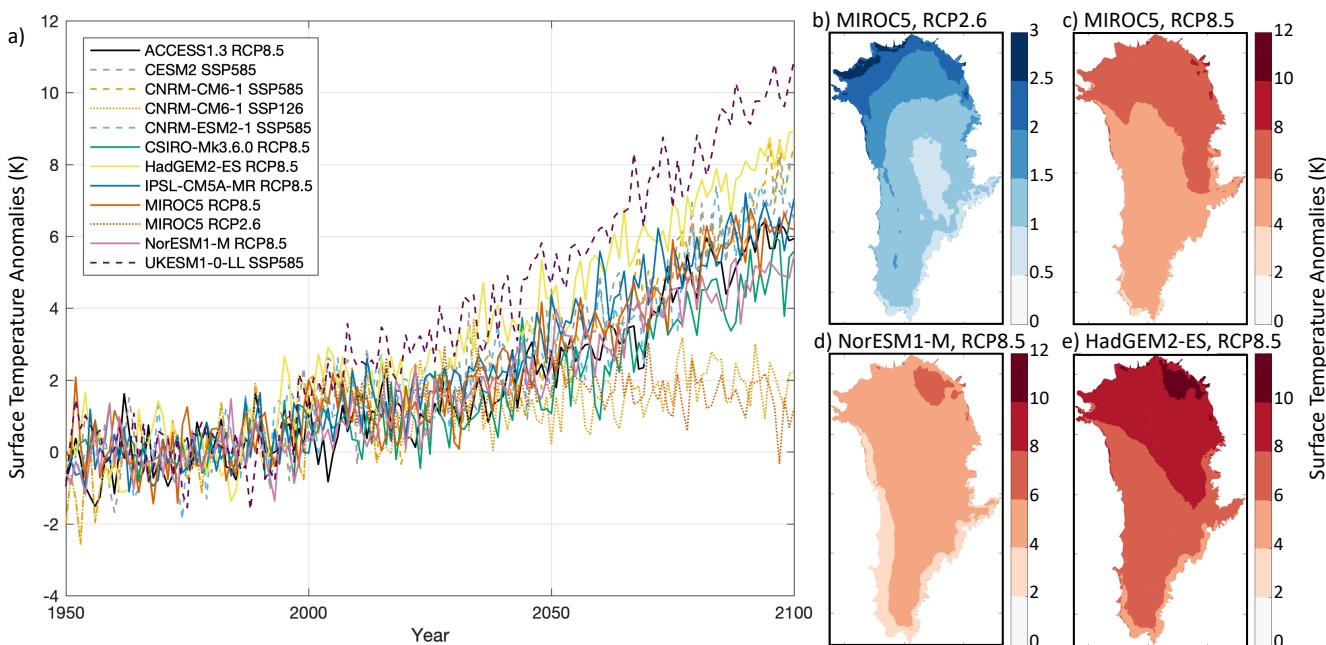

**Figure 4:** Surface temperature anomalies (K) over the Greenland ice sheet under the RCP2.6, RCP8.5, SSP1-2.6 and SSP5-8.5 scenarios. a) Time series of mean surface temperature anomaly from 1950 to 2100 for all AOGCMs and scenarios selected, and (b-e) surface temperature anomaly over the time period 2081-2100 for b) MIROC5 under RCP2.6, c) MIROC5 under RCP8.5, d) NorESM1-M under RCP8.5, and e) HadGEM2-ES under RCP8.5.

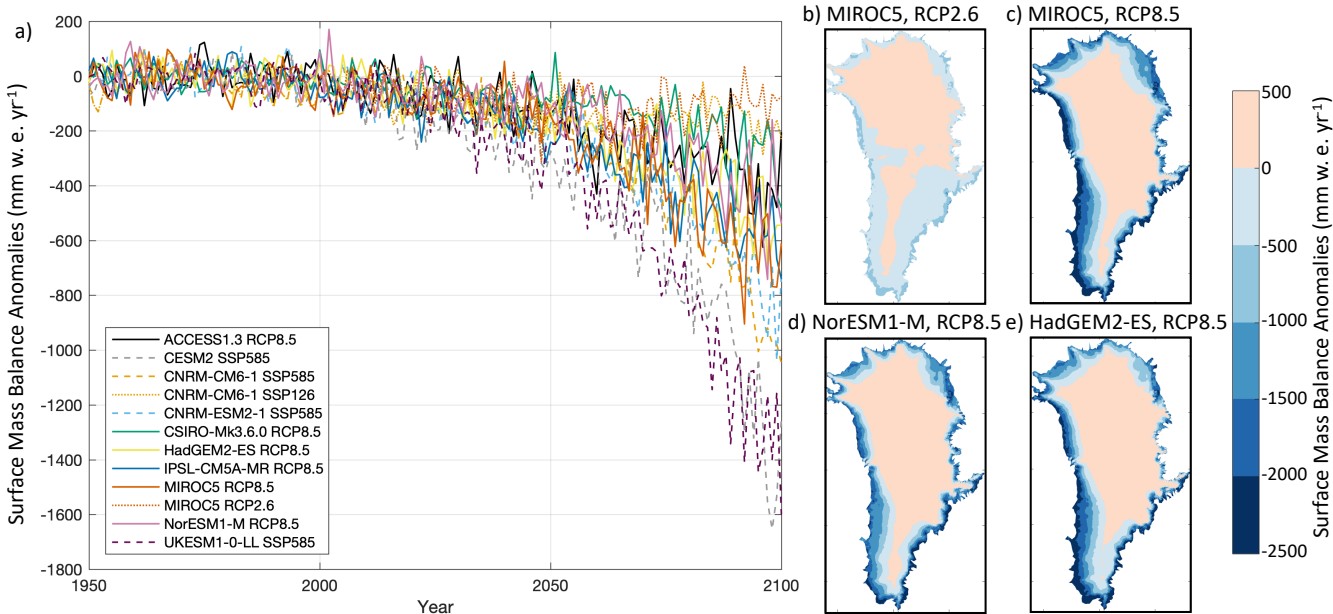

**Figure 5:** Surface mass balance anomalies (mm w. e. yr$^{-1}$) over the Greenland ice sheet under the RCP2.6, RCP8.5, SSP1-2.6 and SSP5-8.5 scenarios. a) Time series of mean surface mass balance anomaly from 1950 to 2100 for all AOGCMs and scenarios selected, and (b-e) mean surface mass balance anomaly over the time period 2081-2100 for b) MIROC5 under RCP2.6, c) MIROC5 under RCP8.5, d) NorESM1-M under RCP8.5, and e) HadGEM2-ES under RCP8.5.

## 5    Oceanic forcing and implementation

Oceanic forcing for standalone ice sheet model simulations consists of temperatures or melt rates at the ice-ocean interfaces (at grounded ice fronts and beneath floating ice shelves), or retreat rate for Greenland models that are not explicitly resolving calving. AOGCM output requires extrapolation under the Antarctic ice shelves and into Greenland fjords, as these regions are not resolved in CMIP models. ISMIP6 protocol allows for ice sheet models to implement their own methods for simulating ice-ocean interactions (*open approach*) using the ISMIP6 extrapolated datasets for oceanic conditions. ISMIP6 also proposes a *standard approach*, where the representation of ice-ocean interactions is specified and datasets that allows for exploring the uncertainty in the standard method are provided. This dual approach of open and standard approach is designed to explore the impact of the range of uncertainty in ice-ocean interactions.

### 5.1.  Antarctica oceanic forcing and implementation

For the Antarctic ice sheet, ISMIP6 provides three-dimensional anomalies of ocean ambient temperature, salinity, and thermal forcing (temperature minus freezing temperature), yearly averaged from 1850 to 2100, for the CMIP simulations listed in Table 1 and Appendix A, which were added to an observational climatology. The observational climatology was produced from a combination of the Marine Mammals Exploring the Oceans from Pole to Pole (MEOP, Roquet et al., 2013, 2014; Treasure et al., 2017), a prerelease of the World Ocean Atlas 2018 (WOA18, Locarnini et al., 2019; Zweng et al., 2019) and the Met Office ENA4 datasets (Good et al., 2013). The CMIP model climatologies, computed over the reference period from Jan 1995 to Dec 2014, are provided for legacy purposes and should not be used in ice sheet parameterizations. Instead, modelers are advised to use anomalies added to the observational climatology. All datasets were extrapolated under ice shelves using an algorithm (described in Jourdain et al. (in review)) that account for sills and troughs, as these bathymetric features affect the flow of oceanic currents. The bathymetry (based on Bedmap2, Fretwell et al., 2013) used for the data preparation, and an example of the resulting sub-ice shelf extrapolation of ocean temperature is shown in Fig 6. The extrapolated datasets are available on a 60 m vertical and 8 km horizontal ISMIP6 Antarctic grid for use by any ice sheet model implementing their own method for prescribing oceanic forcing (*open approach)*. The extrapolation allows for oceanic fields to vary spatially beneath the ice shelves and results, for example, in ambient temperatures that varies with depth.

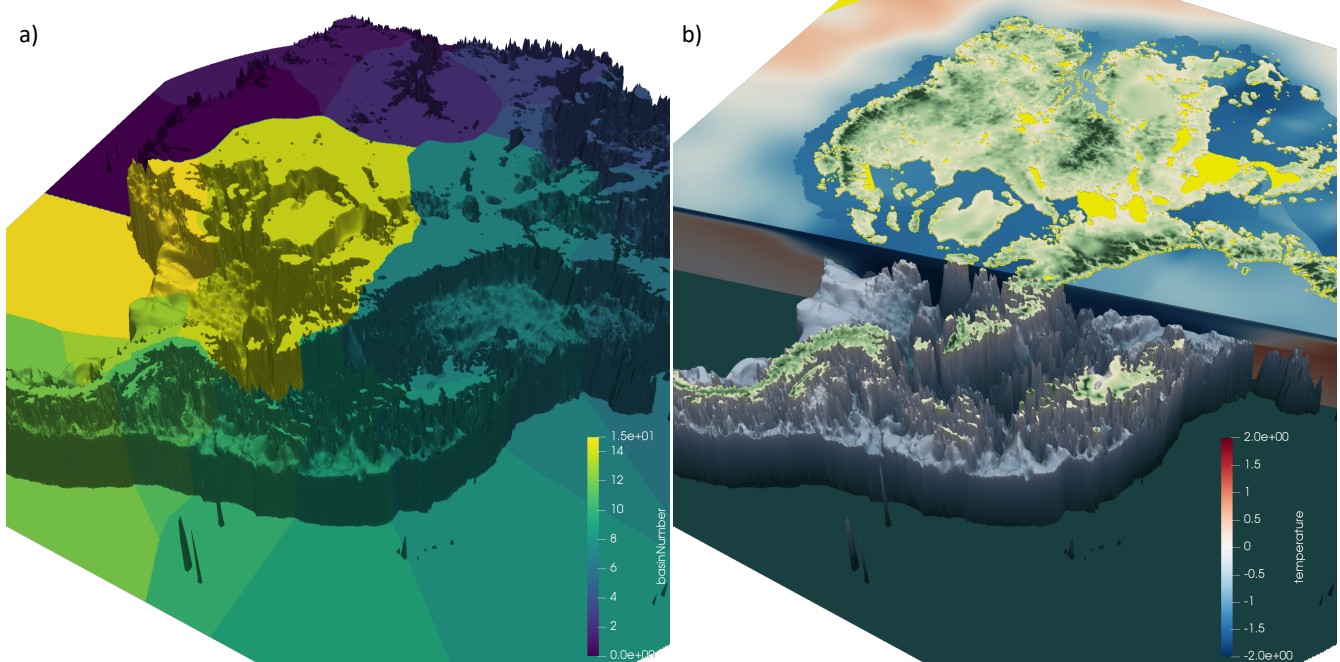

**Figure 6:** a) Bathymetry modified from Fretwell et al. (2013) and IMBIE2 basins (Shepherd et al. 2018) used in the sub-ice shelf extrapolation of oceanic conditions. b) Extrapolated ocean temperature from the observational climatology.

5       For the standard approach, the Antarctic ISMIP6 ocean focus group (Jourdain et al., in review) developed datasets for two sub-ice shelf melt rates parameterizations: non-local quadratic and local quadratic functions of thermal forcing. These parameterizations were evaluated in Favier et al. (2019) for an idealized Pine Island glacier geometry against other commonly used parameterizations, such as the plume parameterization of Lazeroms et al. (2018), the box parameterization of Reese et al. (2018), and a three-dimensional ocean-ice sheet coupled model. The non-local parameterization was found to be in closer agreement with the coupled simulations than the local parameterization and is therefore the preferred approach for ISMIP6 standard simulations. However, the non-local parameterization may be more complex to implement for ice sheet modeling groups as some quantities have to be averaged over regions, which might not be straightforward in large scale parallel models. Ice sheet modeling groups are therefore free to choose between either the non-local or local quadratic parameterizations when participating in the standard experiments.

15       The sub-ice shelf parameterizations take a slightly different form from that proposed by Favier et al. (2019), to allow for regional temperature corrections. The regional sectors are based on the IMBIE2 basins (Shepherd et al., 2018; Mouginot et al., 2017) and extrapolated to the shelf break (Fig. 6 and Jourdain et al. (in review)) to allow for modeled ice sheet extent to be distinct than that of the IMBIE2 observations. The non-local quadratic sub-ice shelf parameterization, $m(x,y)$, takes the form of:

$$m(x,y) = \gamma_0 \left(\frac{\rho_{sw} c_{pw}}{\rho_i L_f}\right)^2 (TF(x,y,z_{\mathrm{draft}}) + \delta T_{\mathrm{sector}})|\langle TF\rangle_{\mathrm{draft}\in\mathrm{sector}} + \delta T_{\mathrm{sector}}| \qquad (7)$$

where $\gamma_0$ is a calibration coefficient, $\rho_{sw}$ and $\rho_i$ are the sea water and ice densities, $c_{pw}$ is the specific heat of sea water, $L_f$ is the fusion latent heat of ice, $TF(x,y,z_{\mathrm{draft}})$ is the thermal forcing at the ice-ocean interface, $\langle TF\rangle_{\mathrm{draft}\in\mathrm{sector}}$ is the thermal forcing averaged over all the ice shelves in a sector, and $\delta T_{\mathrm{sector}}$ is a sector temperature correction. The latter is needed in order to reproduce at the sector scale observation-based melt rate and accounts for biases in observational products. The local quadratic

25  sub-ice shelf melt parameterization takes the form of:

$$m(x,y) = \gamma_0 \left(\frac{\rho_{sw} c_{pw}}{\rho_i L_f}\right)^2 \left(\max[TF(x,y,z_{draft}) + \delta T_{sector}, 0]\right)^2 \qquad (8)$$

As described in Jourdain et al. (in review), both quadratic parameterizations are calibrated against observational estimates using two methods: the "MeanAnt" and the "PIGL" methods. The MeanAnt method calibrates $\gamma_0$ and $\delta T_{sector}$ so that the sub-ice shelf melt parameterizations reproduce the mean Antarctic melt rates of Rignot et al. (2013) and Depoorter et al. (2013). The PIGL method calibrates $\gamma_0$ and $\delta T_{sector}$ so that the sub-ice shelf melt parameterizations reproduce the spatial patterns of melt rates observed close to the grounding line of Pine Island ice shelf (Rignot et al., 2013). To explore the sensitivity of ice sheet simulations to uncertainties in the basal melt rate arising from the calibration, observational melt rates were randomly sampled $10^5$ times to obtain a distribution of possible low ($5^{th}$ percentile), median ($50^{th}$ percentile) and high ($95^{th}$ percentile) values for $\gamma_0$ for both calibration method (MeanAnt and PIGL) and parameterization type (local and non-local). Once $\gamma_0$ had been determined, the median value for $\delta T_{sector}$ was obtained from random sampling ($10^5$ times) of temperature.

To highlight the differences between sub-shelf melt rates, we compare projections under RCP2.6, RCP8.5, SSP1-2.6 and SSP5-8.5 obtained from the non-local sub-shelf parameterizations with the MeanAnt and PIGL calibrations using the median values for $\gamma_0$ and $\delta T_{sector}$. These projections assume that the ice shelf cavities are fixed in time and set to present-day. As described in Jourdain et al. (under review), the average melt rates over 1995-2014 correspond to the observational climatology. The time series of mean cavity sub-shelf melt rate for Pine Island, Thwaites, Ronne-Filchner ice shelves (Fig. 7a and Fig. 8a) project a gradual increase for MeanAnt calibration, with values generally less than twice the present-day conditions by 2100. In contrast, the PIGL calibration project greater melt-rates than their MeanAnt counterpart, as well as larger interannual variability. Both calibrations display regional differences due to the choice of CMIP model. For example, MIROC-ESM-CHEM project negligible change in melt rate over the Ronne-Filchner ice shelves under RCP8.5, but is one of the highest CMIP model for Pine-Island and Thwaites ice shelves with the PIGL calibration. The timing of increasing melt-rate differ between the CMIP models: NorESM1-M and CCSM4 project similar melt-rates with the PIGL calibration in the last three decades of the 21st century for both regions under RCP8.5, but the increase in melt rates from CCSM4 only begins to be significant in the 2060s for the Ronne-Filchner and in the 2040s for Pine Island and Thwaites. HadGEM2-ES projects the largest increase in melt-rate for both calibrations overall under RCP8.5, despite projecting (like most models) negligible change over Ronne-Filchner until the middle of the century. The only model with large Ronne-Filchner melt rate increase prior to the 2050s is UKESM1-0-LL under SSP5-8.5 and PIGL calibration. For the Pine Island and Thwaites ice shelves, the three Tier 1 models MIROC-ESM-CHEM, NorESM1-M, and CCSM4 project end of 21st century melt rates under RCP8.5 that are comparable to that from the SSP5-8.5 models CNRM-ESM2-1 and UKESM1-0-LL, and larger than CNRM-CM6-1 under SSP5-8.5. Over the Ronne-Filchner ice shelf, however, the Tier 1 models NorESM1-M and CCSM4 are closer to CNRM-CM6-1 under SSP5-8.5, and project a third of the melt-rate from CNRM-ESM2-1 and UKESM1-0-LL.

The spatial pattern of mean sub-shelf melt rate over 2081-2100 are illustrated with the Tier 1 simulations listed in Table 2. In the Amundsen Sea sector, the largest melt rates are located over the Pine Island, Thwaites, Crosson and Dotson ice shelves for all scenarios, CMIP models and calibrations (Fig. 7 b-f). Despite similar patterns, the amplitude of the PIGL calibration (Fig. 7f) is an order of magnitude lager than with the MeanAnt calibration (Fig. 7e). The spatial variation in projected melt rate is more apparent over the larger Ronne-Fichner ice shelf for all scenarios, with the melt rate increasing towards the ice shelf-ice sheet junctions, due to the deeper ice-ocean interface (Fig. 8b-f). As illustrated with NorESM1-M, the magnitude of melt rate from the PIGL calibration is again much larger than with the MeanAnt calibration (Fig 8e,f). As expected from the time series, the melt-rates from CCSM4 (Fig. 8d) and NorESM1-M (Fig. 8e) are similar under RCP8.5, and larger than MIROC-ESM-CHEM (Fig. 8c). The latter are also smaller than that from NorESM1-M under RCP2.6 (Fig. 8a).

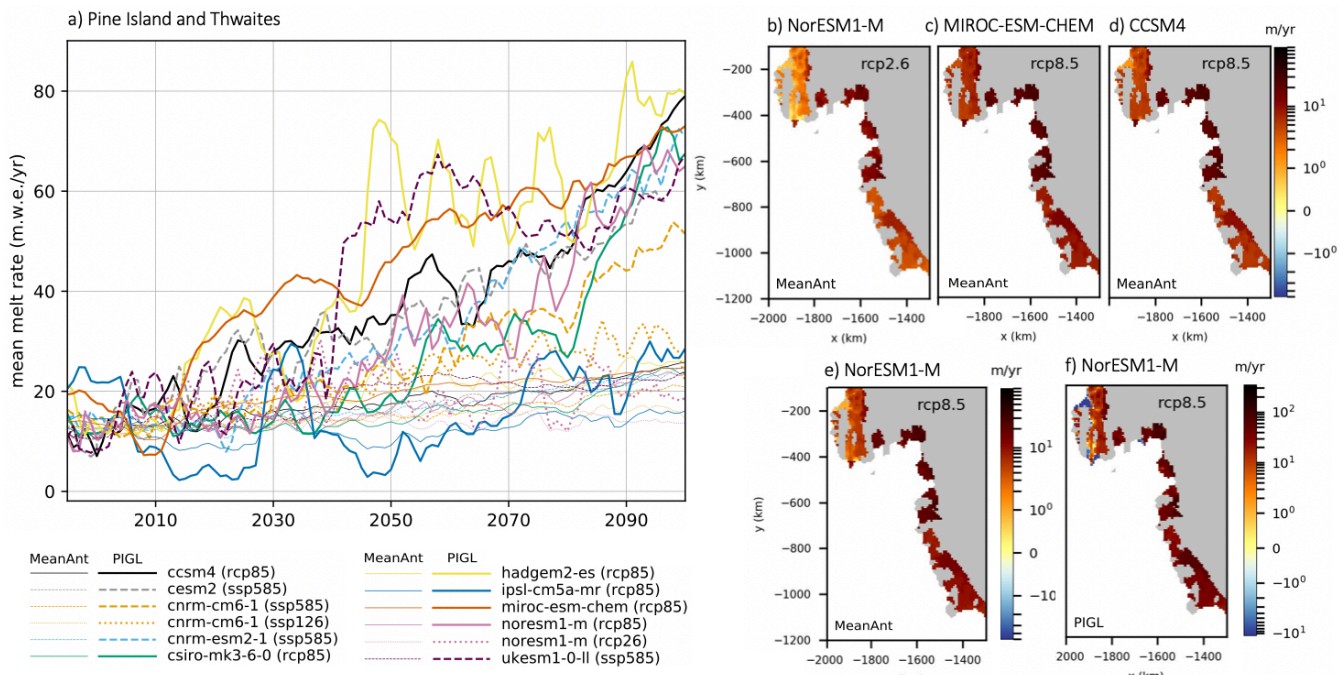

**Figure 7.** Mean cavity basal melt rates (m w.e./yr) under the RCP2.6, RCP8.5, SSP1-2.6 and SSP5-8.5 scenarios over the 21st century for MeanAnt and PIGL calibrations and non-local parameterizations in the Amundsen Sea Sector. a) Time series for Pine Island and Thwaites ice shelves obtained with MeanAnt (thin lines) and PIGL (thick lines) calibrations (adapted from Jourdain et al. (under review)). Spatial patterns of mean sub-shelf basal melt rate from 2081-2100 for the Tier 1 models assuming MeanAnt calibration for b) NorESM1-M under RCP2.6, c) MIROC-ESM-CHEM under RCP8.5, d) CCSM4 under RCP8.5, e) NorESM1-M under RCP8.5, and PIGL calibration for f) NorESM1-M under RCP8.5. The projections from the MeanAnt and PIGL calibrations assume that the ice shelf cavities have not evolved and are set to present day.

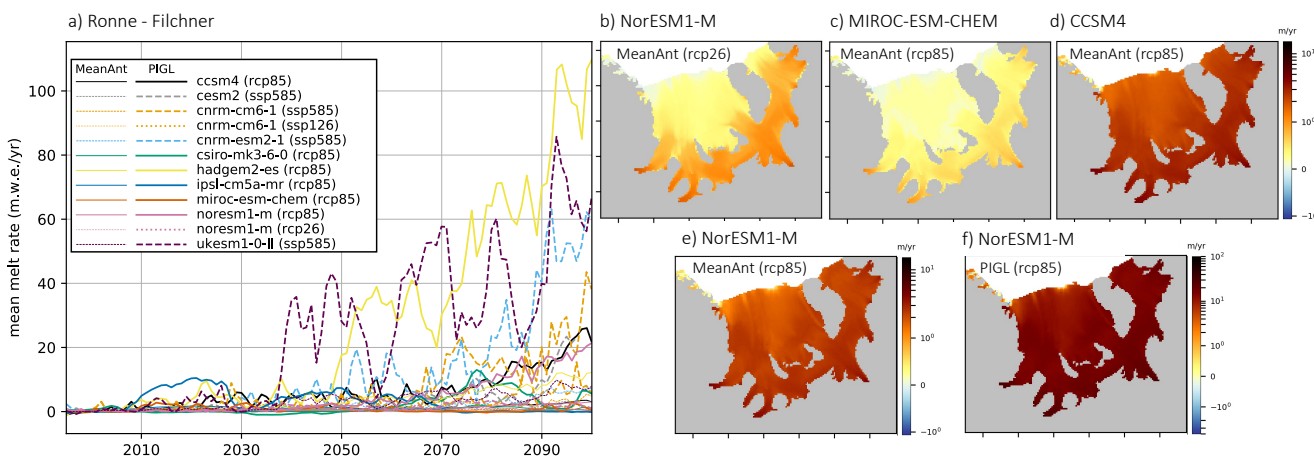

**Figure 8.** Mean cavity basal melt rates (m w.e./yr) under the RCP2.6, RCP8.5, SSP1-2.6 and SSP5-8.5 scenarios over the 21st century for MeanAnt and PIGL calibrations and non-local parameterizations in the Weddell Sea Sector. a) Time series for Ronne-Filchner ice shelves obtained with MeanAnt (thin lines) and PIGL (thick lines) calibrations (adapted from Jourdain et al. (under review)). Spatial patterns of mean sub-shelf basal melt rate from 2081-2100 for the Tier 1 models assuming MeanAnt calibration for b) NorESM1-M under RCP2.6, c) MIROC-ESM-CHEM under RCP8.5, d) CCSM4 under RCP8.5., e) NorESM1-M under RCP8.5, and PIGL calibration for f) NorESM1-M under RCP8.5. The projections from the MeanAnt and PIGL calibrations assume that the ice shelf cavities have not evolved and are set to present day.

## 5.2. Greenland oceanic forcing and implementation

The ISMIP6 Greenland ocean focus group (Slater et al., 2019, 2020) proposed two methods for implementing oceanic forcing: the "retreat implementation" and the "submarine melt implementation." The two approaches are needed in order to maximize participation from current state-of-the-art ice sheet models. The retreat implementation prescribes the temporal evolution of ice extent using masks that specify annual ice sheet extent. Both implementations rely on yearly average datasets of subglacial discharge per glacier and two-dimensional ocean thermal forcing (temperature minus freezing temperature), varying horizontally but not with depth, from 1950 to 2100 for the CMIP simulations listed in Table 3 and Appendix A. In the retreat implementation, these datasets are used to create projections of marine-terminating glacier retreat from 2015 to 2100. In the submarine melt implementation, these datasets allow ice sheet models to calculate submarine melt rate and combine with a calving rate to obtain total frontal ablation. Here we provide an overview of the ocean forcing strategy for Greenland ice sheet models taking part in ISMIP6. For details of the retreat parameterization readers are referred to Slater et al. (2019) and full details of the two implementations can be found in Slater et al. (2020).

Subglacial discharge is freshwater that emerges from beneath the ice into proglacial fjords at marine-terminating glacier termini. The discharge is a result of surface meltwater runoff that reaches the base of the ice sheet and is routed through the subglacial hydrological system prior to reaching the ice margin. Because many ice sheet models do not include a physical representation of the surface processes that generate runoff, nor do they simulate evolving subglacial hydrology, the ISMIP6 Greenland ocean focus group (Slater et al., 2020) recommends approximating subglacial discharge as a spatial aggregation of surface runoff, calculated by a regional climate model, over a static delineation of each glacier's subglacial hydrological catchment. In other words, it is assumed that surface meltwater runoff generated within a particular catchment is instantaneously transported to the ice sheet bed and routed to the terminus, according to water routing that remains constant over the projection time period. As described in Slater et al. (2020), the annual subglacial discharge dataset was produced from downscaled MAR surface runoff (Section 4.2), with subglacial hydrological catchments delineated using ice sheet geometry corresponding to present day surface elevation (Howat et al., 2014) and basal topography (BedMachine3; Morlighem et al., 2017). In addition, because the CMIP AOGCMs used as forcing in MAR may differ from present-day climate, the MAR surface runoff was bias corrected so that it better matches present-day surface runoff for each glacier from the time period between 1995 to 2014. This was done using surface runoff estimates from the RACMO2.3p2 regional climate model, which is forced by ERA-Interim atmospheric reanalysis (Noël et al., 2018), as a measure of the present-day runoff values (Slater et al., 2020). Ice sheet models with a configuration that differs substantially from these observational datasets should consider computing their own subglacial discharge dataset using the appropriate surface runoff and water routing scheme. This guidance applies to groups that use the SMB remapping technique described in Section 4.2.

As described in Slater et al. (2020), the CMIP oceanic datasets have been extrapolated into fjords for use by groups choosing the submarine melt implementation or in a model's own scheme. The extrapolation takes into account the ocean bathymetry and subglacial topography following the method of Morlighem et al. (2019). The technique identifies an "effective depth", which is the deepest point within a fjord connected with the open ocean. The method then assumes that ocean mass shallower than the effective depth is in contact with the open ocean, while water deeper than the effective depth is sheltered from the open ocean. For depths shallower than the effective depth the temperature and salinity are set to the closest ocean conditions at that particular depth, whereas for deeper regions the temperature and salinity are set to the values corresponding to the effective depth in that region. In addition, both salinity and temperatures have been bias corrected with present-day observations from the Hadley Centre EN4.2.1 dataset (Good et al., 2013). Note that in the preparation of the retreat implementation datasets (Slater et al., 2019), bathymetry was not taken into account, instead oceanic properties were averaged over the seven sectors between 200 and 500 m.

In the standard approach, the retreat implementation is designed to be simple enough to be implemented by most ice sheet models (Slater et al., 2019; 2020). The approach follows the parameterization of Cowton et al. (2018) and combines anomalies in subglacial runoff and ocean thermal forcing to produce terminus retreat rates that are implemented via the use of a time-variable ice mask. As ice sheet models may not capture small outlet glaciers and may have distinct location for individual outlet glaciers compared to the observations, it is not practical to provide a dataset for each observed outlet glacier. Instead, separate retreats are provided for seven ice-ocean sectors:

$$dL = \kappa \, d(Q^{0.4}TF) \tag{9}$$

where $dL$ is the retreat distance of each glacier in a particular sector, $\kappa$ is a calibration constant described in Slater et al. (2019), and $Q$ is the summer subglacial runoff generated from the mean of June, July and August. The uncertainty associated with the retreat implementation is investigated via low, median, and high retreat scenarios for each CMIP model. These were obtained from an ensemble of $10^4$ ice-flux weighted trajectories for each ice-ocean sector, and set to the 25th percentile (low), 50th percentile (median) and 75th percentile (high) for each CMIP model. Implementation for a specific ice sheet model requires the model's initial ice mask and the observed basal topography, in order to identify ice prone to outlet glacier retreat. For each model participating in ISMIP6 and experiment, a specific time-variable retreat mask was produced using the method described in Appendix D. These annual "land_ice_area_fraction" masks cover the period from 2015 to 2100. The mask values are set to 0.0 over ice free regions, to 1.0 over fully ice-covered regions and values in between for grid cells that are partially covered. Ice sheet models should apply full retreat for mask values set to 0.0, no retreat for mask set to 1.0 and partial retreat for mask values in between 0.0 and 1.0. The retreat datasets are not optimum for glaciers that have a floating ice shelf, as the datasets were calibrated using the ice front position of glaciers that do not have ice shelves. Nonetheless, for the few Greenland glaciers with floating ice shelves, it is suggested that the retreat be imposed at the ice front, rather than the grounding line. Groups that are able to compute sub-ice shelf melt may use the basin thermal forcing datasets provided for each basin.

In the submarine melt implementation, the melt rate and calving rate along each marine-terminating glacier's terminus must be calculated by the ice sheet models and combined to determine the frontal ablation rate. Previous work has shown that submarine melt rate can be parameterized as a function of ocean thermal forcing and subglacial discharge (Xu et al., 2013; Rignot et al., 2016) and we utilize this same parameterization:

$$\dot{m} = (3 \times 10^{-4} h q^{0.39} + 0.15)TF^{1.18} \tag{10}$$

where $\dot{m}$ is the submarine melt rate, $h$ is the depth at the grounding line, $q$ is the annual subglacial runoff and $TF$ is the ocean thermal forcing. The melt rate is the average rate of ice melt across the entire terminus. Thus, at a given ice sheet model time step, $q$ and $TF$ can be sampled from the ISMIP6 forcing files at the location of each marine-terminating glacier terminus and the calculated melt rate applied across each terminus face. Each group is free to simulate calving using any approach they see fit; the summation of melt and calving then provides the ice sheet model with total frontal ice ablation. This method may require substantial model development and a high model resolution of the outlet glaciers, but, whereas the retreat implementation imposes the same retreat amount for all glaciers within each sector, the submarine melt implementation allows for adjacent glacier to retreat at a different rate, since the melt rate is computed for each individual marine-terminating outlet glacier. Ice sheet models that already have a different submarine melt parameterization are encouraged to implement the Rignot et al. (2016) parameterization as well. The difference between their existing melt parameterization (open experiment) and the proposed standard submarine melt implementation provides insight into the uncertainty in frontal melt parameterizations and the impact of the resulting ocean forcing on ice sheet evolution. At the same time, the possible wide range in submarine melt arising from different parameterizations obfuscates our understanding of how climate forcing uncertainty propagates into submarine melt uncertainty. Thus, the proposed implementation of a standard submarine melt parameterization allows us to examine how the spread in climate projections drives the spread in submarine melt rates and glacier retreat.

To illustrate the difference in the two ocean forcing approaches, we compare retreat against melt rates for two neighboring glaciers chosen at random and focus on the Tier 1 simulations under RCP2.6 and RCP8.5. In the retreat

implementation, these two glaciers will retreat over the same distance in any given AOGCM and RCP scenario because all glaciers in this sector are forced with the same rate of terminus retreat (Fig. 9). However, these two glaciers exhibit very different projected melt-rate anomalies, defined as the difference in mean melt rates between 2081-2100 and 1995-2014 (Fig. 10). Kong Oscar Gletsjer (glacier 51) will undergo a substantial but varied increase in melt rates in most RCP8.5 projections (Fig. 10a), whereas glacier 53 shows modest increases in melt rates in all projections (Fig. 10a). Thus, the ocean forcing in the melt implementation will be stronger for Kong Oscar Gletsjer than glacier 53. The melt implementation allows for greater spatial variability compared to the retreat implementation. The melt implementation also allows for greater variability for a given glacier as its terminus retreats, and feedback with bedrock topography, e.g., higher melt rates where the bedrock is deeper (Fig. 10 b-e). For a given glacier, the spatial patterns of projected melt rate anomalies are similar between the different CMIP models, but the magnitude differs as a result of different thermal ocean forcing and subglacial discharge. For the retreat implementation, the choice of a CMIP model impacts the timing and location of ice retreat (Fig. 9 b-e), with NorESM1-M resulting in earlier retreat for a given location, and the retreat reaches further inland by 2100 compared to the other CMIP models. Nonetheless, both the retreat implementation and melt implementation show consistencies with each other (Fig. 9a and Fig. 10a): higher ocean forcing is projected for NorESM1-M, followed by MIROC5 a under RCP8.5. HadGEM2-ES project half of the retreat-rate and melt-rate at the end of the 21$^{st}$ century compared to NorESM1-M for this region and climate scenario.

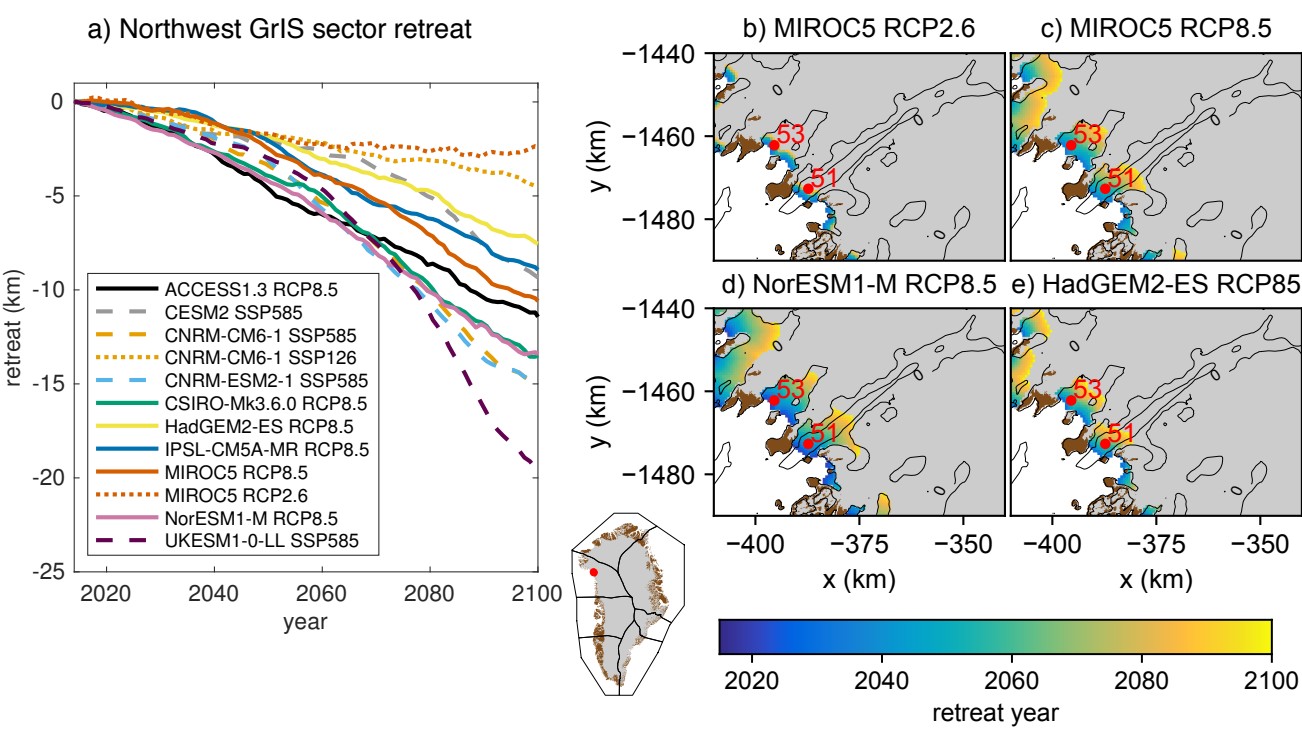

**Figure 9:** Retreat for selected Greenland ice sheet glaciers under the RCP2.6, RCP8.5, SSP1-2.6 and SSP5-8.5 scenarios. a) Time series of retreat over the 21$^{st}$ century for the Northwest sector (adapted from Slater et al., 2019) and retreat from 2015 to 2100 for b) MIROC5 under RCP2.6, c) MIROC5 under RCP8.5, d) NorESM1-M under RCP8.5, and e) HadGEM2-ES under RCP8.5. The bedrock contour from Morlighem et al. (2017) is shown as black lines, gray indicates ice sheet mask. Red numbers indicate Kong Oscar Gletzer (51) and unnamed glacier (53) discussed in the text.

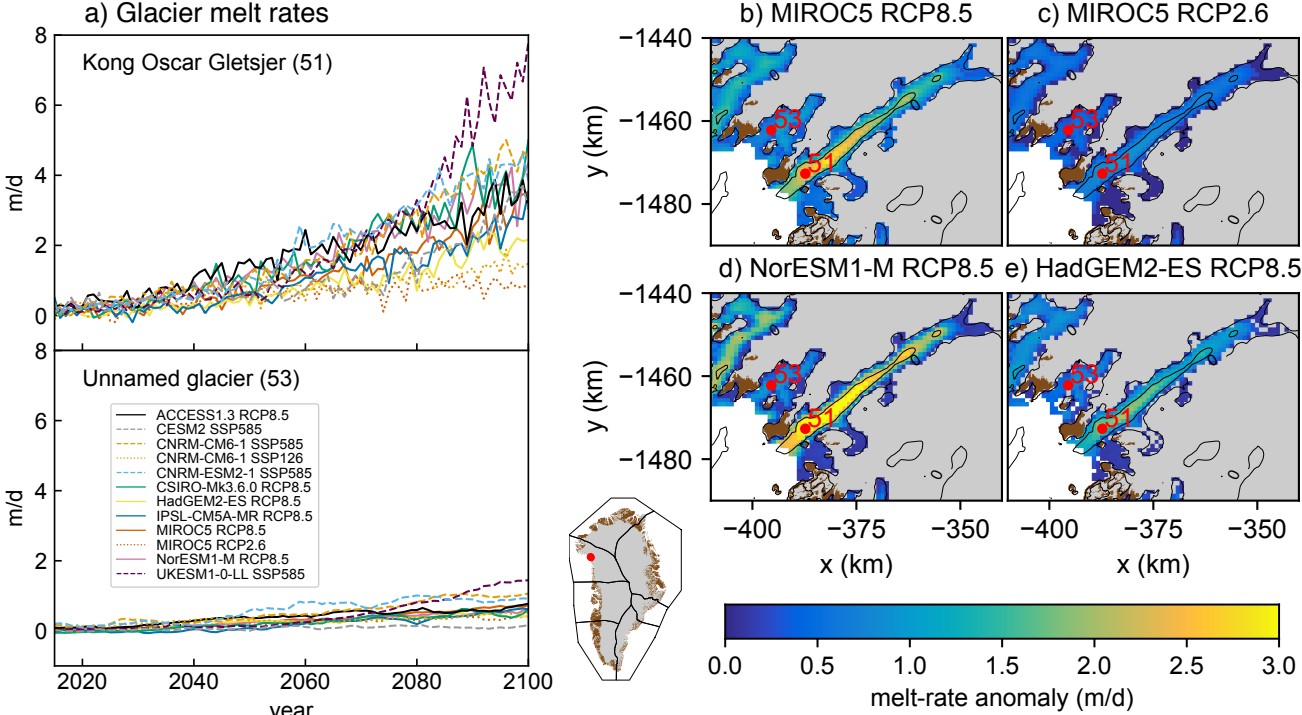

**Figure 10:** Melt rate anomalies (m d$^{-1}$) for selected Greenland ice sheet glaciers under the RCP2.6, RCP8.5, SSP1-2.6, and SSP5-8.5 scenarios. a) Time series of mean melt rate anomaly over the 21$^{st}$ century and mean melt rate anomaly over the time period 2081-2100 for b) MIROC5 under RCP2.6, c) MIROC5 under RCP8.5, d) NorESM1-M under RCP8.5, and e) HadGEM2-ES under RCP8.5. The bedrock contour from Morlighem et al. (2017) is shown as black lines, gray indicates ice sheet mask. Red numbers indicate Kong Oscar Gletzer (51) and unnamed glacier (53) discussed in the text. Note that for HadGEM2-ES missing pixels are due to bias correction making ocean thermal forcing in some of the cooler models slightly negative, resulting in no melt-rate anomaly. The calculation of melt-rate anomaly assumes that the ice sheet thickness and terminus position does not evolve with time.

## 6   Antarctic ice shelf fracture

The 2002 collapse of the Larsen B ice shelf was attributed to enhanced surface melting (Sergenkio and MacAyeal, 2005; van den Broeke, 2015). Enhanced surface melt and water ponding can then trigger hydrofacturing and ice shelf collapse (Vaughan and Doake, 1996; Scambos et al., 2000; Scambos et al., 2009). Other mechanisms associated with ice shelf collapse include rheological weakening, ocean waves, surface load changes (MacAyeal et al., 2003; Braun and Humbert, 2009, Borstad et al., 2013). It has been hypothesized that ice cliffs resulting from ice shelf collapse could themselves become inherently unstable, a process called marine ice cliff instability (Bassis and Walker, 2011; DeConto and Pollard, 2016). As the processes for ice shelf collapse remain poorly understood and are rarely implemented in continental ice sheet models, ISMIP6 focusses on ice shelf collapse due to enhanced surface melt and provides time varying mask for ice shelf fracture. The objective is to investigate the impact of ice shelf collapse, and the resulting loss of buttressing, by performing similar experiments without and with ice shelf collapse. The datasets were prepared following the method described in Trusel et al. (2015), which derives an annual surface melt from CMIP near surface temperatures:

$$M_{AOGCM}(x,y,t) = 1183 \times e^{\left(0.4557 \times T2m_{AOGCM,adjusted}(x,y,t)\right)}$$

where $M_{AOGCM}(x,y,t)$ is the annual CMIP derived surface melt flux (in mm w.e. yr$^{-1}$) and $T2m_{AOGCM,adjusted}(x,y,t)$ is the downscaled, bias-corrected, CMIP near surface temperature. Ice shelves are assumed to collapse following a 10 consecutive year period with annual melt above 775 mm w.e. yr$^{-1}$, a threshold suggested by Trusel et al. (2015). ISMIP6 provides annual

masks of ice shelf collapse covering 1995 to 2100 for CCSM4, CSIRO-Mk3.6.0, HadGEM2-ES, IPSL-CM5A-MR, MIROC-ESM-CHEM, and NorESM1-M under RCP8.5 (Fig. 12). Ice shelf collapse masks were created on the 4 km ISMIP6 Antarctica grid, and these were conservative interpolated to generate masks at 2, 8, 16 and 32 km resolutions. Modelers should use the grid that is the most appropriate for their models. In the 4 km dataset, the mask values are set to 1.0 when the ice shelf is prone to collapse and to 0.0 for no collapse. The interpolation to coarser dataset may result in fractional mask values, with a value between 0.0 and 1.0 indicating partial collapse. Ice sheet models should remove the ice shelf when the mask is set to values greater than 0.0. If flagged, collapse is assumed to occur on January 1st of each year. A modeled ice shelf extent may not always exactly correspond to an ice shelf in the collapse masks dataset. In the event where the ice sheet model considers the ice to be grounded, but the ice shelf collapse mask indicates ice shelf removal, the collapse mask should not be imposed. Application of the mask may also result in ice shelf regions that are now detached from the ice shelf. In this case, these "iceberg" should be removed as well. Finally, a collapsed ice shelf should not regrow. The ice flow response to collapsed ice shelf is left at the discretion of modeling groups.

As indicated in Table 2, the Tier 1 model for this experiment is CCSM4. This model was prioritized over the two other models selected as part of the core experiments (MIROC-ESM-CHEM and NorESM1-M) because of its largest projected ice shelf collapse area under RCP8.5 (Fig. 11a). In CCSM4, ice shelf collapse is initially sporadic, with a collapse in the mid 2020s that last until the early 2030s. The decades of the 2030s and 2040s do not experience collapse. The second period of ice shelf collapse begins in the mid 2050s, lasts approximately three decades and impacts ice shelves on both sides of the Peninsula. The last, more rapid, phase of collapse begins in the early 2080s and impacts ice shelves in all regions around Antarctica. The spatial map of ice shelf collapse (Fig. 11b) indicates that the early collapse occurs predominantly at the fringes of the ice shelves in the Peninsula, and in particular over the Wilkins ice shelf. The Larsen ice shelves and the George VI ice shelf collapse during the second period and are gone in 2070. The final phase of collapse, depicted by the conditions in 2090, includes the Abbot ice shelf and fringes of the Getz ice shelf, in West Antarctica, but also occurs at the terminus of small ice shelves throughout the East Antarctic. The large Ronne-Filchner and Ross ice shelves are not projected to experience ice shelf collapse, nor are the ice shelves fed by Pine Island and Thwaites glaciers. This "step wise" increase over the 21st century in the projected ice shelf area is seen in all the datasets, however the timing, duration and area affected differ between the models. The projected ice shelf collapse from HadGEM2-ES from 2080 to 2100 follows closely that from CCSM4 in terms of total area affected, but HadGEM2-ES projects a smaller area and delayed collapse for ice shelves on both side of the Peninsula and the Abbot ice shelf (Fig. 11a,c), comparable timing and extent for the Getz ice shelf, and a larger number of East Antarctic ice shelves at the end of the century. The projected behaviors for MIROC-ESM-CHEM and CSIRO-Mk3.6.0 converge in the period 2080 to 2100, despite MIROC-ESM-CHEM trailing behind initially. MIROC-ESM-CHEM project some collapse over the fringes of the East Antarctica ice shelves towards the end of the century, while CSIRO-Mk3.6.0 does not, explaining the greater total ice shelf area collapse for MIROC-ESM-CHEM at the end of the century. In contrast, both NorESM1-M and IPSL-CM5A-MR only project ice shelf collapse in the Peninsula (Fig. 11d,f), with IPSL-CM5A-MR being the model with the smallest area affected (Fig. 11a). As none of the selected CMIP5 models project ice shelf collapse over the Pine Island and Thwaites ice shelves, we do not expect this suite of experiments to trigger a collapse of the West Antarctic ice sheet. The ice shelf loss is concentrated in the Peninsula for all models, but the timing and areas affected differ between the them. Therefore, the Antarctic wide projected sea level change from this implementation of ice shelf triggered hydrofracture is anticipated to be small.

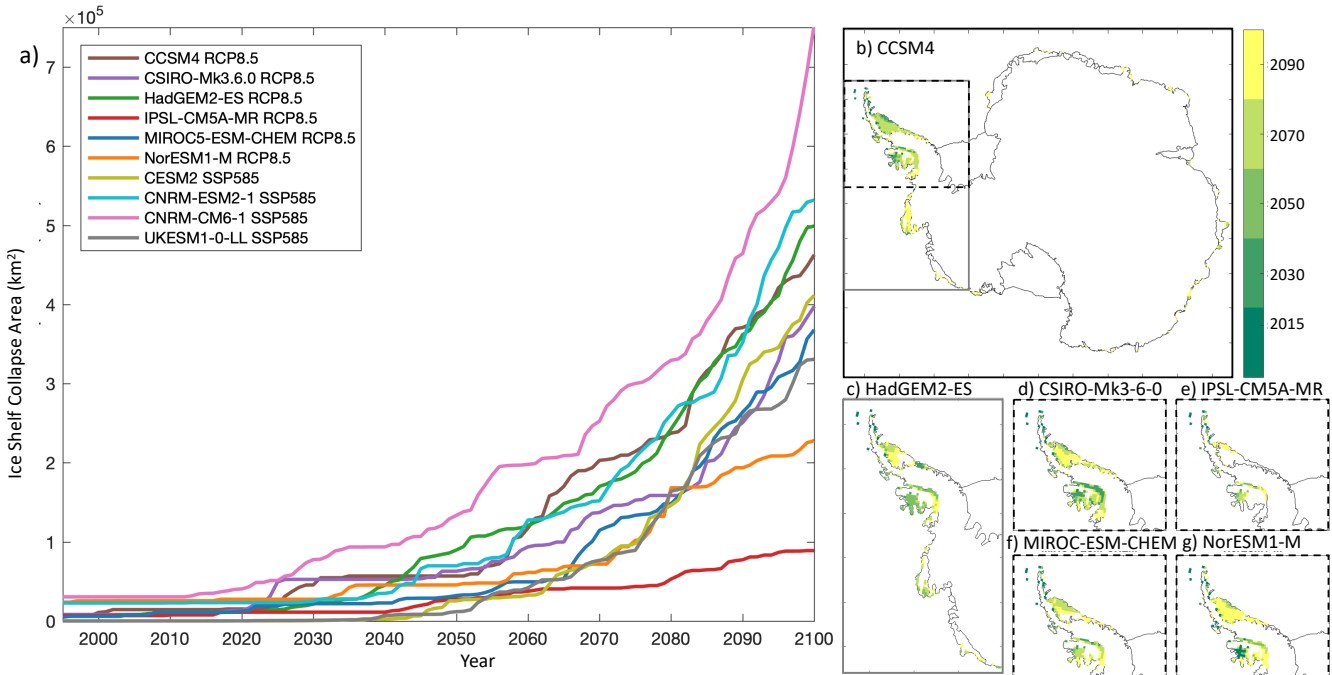

**Figure 11:** Time series of ice shelf collapse area under the RCP8.5 and SSP5-8.5 scenarios over the 21st century a) and corresponding ice shelf collapse masks spatial evolution b-g). The CMIP5 models are CCSM4 (blue, b), HadGEM2-ES (green, c), CSIRO-Mk3.6.0 (green, d), IPSL-CM5A-MR (cyan, e), MIROC-ESM-CHEM (red, f), and NorESM1-M (pink, g). The masks spatial extent shown originate from the 4km datasets and correspond to years 2015, 2030, 2050, 2070 and 2090.

## 7   Discussion and conclusion

The protocols presented in this paper differ in detail between Greenland and Antarctica because the key processes by which ice is lost is different for each ice sheet. Certain aspects of the protocol are also influenced by the time limitations faced by ISMIP6 (due to the delay in CMIP6 simulations) and computational constraints limiting the use of RCM to Greenland only. Both atmospheric and oceanic forcings are required for both ice sheets, presented to the ice-sheet model as an evolving flux of ice entering or leaving the ice sheet, or as fraction of ice that should be removed. Examples include the balance between snow accumulation and meltwater runoff on the ice sheet's surface (SMB), melt and/or refreezing on the underside of floating ice shelves, melt from the vertical faces on marine-terminating outlet glaciers and the calving of icebergs (both from marine-terminating outlet glaciers and in the collapse of floating ice shelves). Several factors determine the choice of how this mass flux is calculated. These include the scale over which the mass flux varies spatially compared to the native resolution of AOGCMs. For instance, steep topographic gradients at the edge of the ice sheet introduce sharp spatial gradients in SMB, or the need to resolve runoff over the narrow Antarctic ice shelves, that are crucial in determining ice-sheet response but are typically not resolved by AOGCMs. Other factors influence the quality of the SMB derived from AOGCMs besides resolution, such as the choice of physical parameterizations (e.g. Palerme et al., 2017) or limitation in the processes included in snowpack models, for example. In some cases, the key mass fluxes may not be determined by the AOGCMs at all. This is true, for instance, of melt from the underside of ice shelves and from Greenland marine-terminating outlet glaciers, or the feedback of the ice sheet's evolving geometry on the mass fluxes themselves.

In general, one of three approaches is used to determine each mass flux. Firstly, in some cases it is possible to employ the mass flux determined within the AOGCM directly. Second, an RCM can be used to simulate the necessary mass fluxes. This approach is used to provide SMB forcing for Greenland whereby the MAR RCM is forced using lateral and surface boundary conditions from the relevant AOGCMs (see Fettweis et al., 2017; Hofer et al. under review). The SMB forcing from

Antarctica was not chosen to be obtained from an RCM due in part to the considerable time and computational resources required with this approach. Finally, a parametrization can be used to create the relevant mass flux based on variations in the relevant climate variables supplied by AOGCMs. This approach is used in the case of iceberg calving and melt from the underside of Antarctic ice shelves and Greenland marine-terminating outlet glaciers. Note that in addition to the primary forcing obtained from the AOGCM (for instance, an index of subsurface ocean temperature for ice-shelf melt) these parameterizations may also require ancillary information; for example, the impact of surface runoff on the frontal melt of marine-terminating outlet glaciers in Greenland. In some instances, climatic forcing cannot be imposed directly as a mass flux, and instead is implemented as a removal of ice. This is the case for the retreat rate implementation of Greenland outlet glaciers and for the fracture of Antarctic ice shelves due to increased surface melt.

The ISMIP6 experimental protocol attempts to balance the need for a full exploitation of the various sources of uncertainty on sea level projections against the high computational costs of some of the models likely to particulate in the intercomparison. Four types of uncertainty are considered. The first is the choice of emission scenarios – we chose to focus primarily on the high-end RCP8.5 scenario in addition to a limited amount of work with the RCP2.6 to bracket that uncertainty associated with emissions. The second form of uncertainty is the choice of the AOGCM within the overall CMIP ensemble for a given emission scenario. Here, a great deal of care was used in selecting AOGCMs that represent the range of climate projections within the ensemble, as well as simulating the present-day climate of the ice sheets adequately. Issues related to delays within the CMIP6 led to a focus on CMIP5 supplemented by available CMIP6 results. This has the advantage that existing work on understanding variations within the CMIP5 ensemble (Agosta et al., 2015) could be used to aid model selection. Please see Barthel et al. (2020) for a detailed description of the procedures employed. The third source of uncertainty is that associated with the ice-sheet models themselves. This includes the physics used by individual models (structural uncertainty) and the values chosen for poorly constrained parameters within the model (parametric uncertainty). These types of model uncertainty can be difficult to explore, but there are several ways in which ISMIP6 aims to do this. Firstly, the "standard" experiments, described by the protocols presented here and required from each participant, allow for direct comparison between ISMs. These models can vary widely in ice flow physics and numerical techniques – and therefore comparing the outcome of the standard experiments from different models, when all other experimental choices are equal, helps quantify the structural uncertainty. Secondly, modelling groups are also encouraged to submit "open" experiments in which they can employ their own parameterisations and include physical processes which may not be included in the standard experiments (which were designed to be compatible with all the participating models). One form of uncertainty that ISMIP6 does not sample in a systematic manner is ice sheet model parameter uncertainty. This requires quantifying the spread in projections by constructing an ensemble of simulations with model parameters varied in a methodical way as it would be computationally prohibitive for many of the participating ISMs. Modelling groups are however encouraged to repeat the experiments with different versions of their models. The final source of uncertainty is linked to our current lack of understanding of ice-ocean interactions: some of the ISMIP6 forcings that are used to parameterize ice-ocean processes in the ice sheet models have been provided with a range of values, to reflect uncertainty in the associated parameterizations.

Experiments are separated into "core" (or Tier 1) experiments that every participant is expected to complete, and optional "targeted" experiments (Tiers 2 and Tier 3). This approach allows participants with models with faster runtimes to explore particular sources of uncertainty in more detail, while allowing all models to be compared across smaller set of core experiments. The experiments are also separated to allow comparison between projections in which only one type of uncertainty changed. For instance, holding parameter and CMIP model choice fixed, to allow the impact of emission scenarios to be isolated. CMIP model selection results in a range of future climate scenarios, with some CMIP models projecting a warmer atmosphere but colder ocean conditions, and vice versa. As illustrated in this paper, there are strong regional differences in the atmospheric and oceanic forcings, which will transfer to differences in ice flow response and projected sea-level. The datasets presented in this manuscript were specifically prepared by ISMIP6 and are available for community use.

The preparation of this ISMIP6 protocol for standalone ice sheet model brought together multiple communities towards the goal of understanding the uncertainty in future sea level change. ISMIP6 simulations and scientific achievements are expected to support the WCRP Grand Science Challenges on "Melting Ice and Global Consequences" and "Regional Sea-level Change and Coastal Impacts", and contribute to the projected sea-level change in the IPCC Sixth Assessment Report.

We anticipate that our protocol will benefit sea-level research through the design of a consistent set of ISMIP6 simulations which will provide a basis for projections, comparative analysis, and further targeted ice sheet modelling activities. Progress towards resolving the pressing scientific and practical problem of projecting the contribution to global sea-level change from the ice sheets, and reducing its uncertainty, requires continued collaboration between multiple disciplines in the Earth-system sciences. Towards this end, we hope that ISMIP6 will succeed in promoting the scientific study of ice sheets and climate as a

coupled system.

**Appendix A: ISMIP6 Tier 2 and Tier 3 experiments**

The optional Tier 2 and Tier 3 ISMIP6 experiments are described in Tables A1-A4 for Antarctica and Tables A5-A6 for Greenland. These experiments complement the mandatory Tier 1 experiments. Groups are highly encouraged to perform Tier

2 experiments, which complete the CMIP5 models selected in Barthel et al. (2020) for ISMIP6. In contrast, the CMIP6 models were a selection of opportunity. Tier 3 experiments consist of experiments with atmosphere or ocean only forcing, or the remainder of the ice-shelf hydrofracture experiments. Groups can choose to focus on one aspect of these experiments.

**Table A1:** Description of ISMIP6 Antarctica Tier 2 simulations

| Experiment ID | Scenario | CMIP AOGCM | Standard/Open | Ocean forcing | Fracture |
|---|---|---|---|---|---|
| expA1 | RCP8.5 | HadGEM2-ES | Open | Medium | No |
| expA2 | RCP8.5 | CSIRO-Mk3.6.0 | Open | Medium | No |
| expA3 | RCP8.5 | IPSL-CM5A-MR | Open | Medium | No |
| expA4 | RCP2.6 | IPSL-CM5A-MR | Open | Medium | No |
| expA5 | RCP8.5 | HadGEM2-ES | Standard | Medium | No |
| expA6 | RCP8.5 | CSIRO-Mk3.6.0 | Standard | Medium | No |
| expA7 | RCP8.5 | IPSL-CM5A-MR | Standard | Medium | No |
| expA8 | RCP2.6 | IPSL-CM5A-MR | Standard | Medium | No |
| expB1 | SSP5-8.5 | CNRM-CM6-1 | Open | Medium | No |
| expB2 | SSP1-2.6 | CNRM-CM6-1 | Open | Medium | No |
| expB3 | SSP5-8.5 | UKESM1-0-LL | Open | Medium | No |
| expB4 | SSP5-8.5 | CESM2 | Open | Medium | No |
| expB5 | SSP5-8.5 | CNRM-ESM2-1 | Open | Medium | No |
| expB6 | SSP5-8.5 | CNRM-CM6-1 | Standard | Medium | No |
| expB7 | SSP1-2.6 | CNRM-CM6-1 | Standard | Medium | No |
| expB8 | SSP5-8.5 | UKESM1-0-LL | Standard | Medium | No |
| expB9 | SSP5-8.5 | CESM2 | Standard | Medium | No |
| expB10 | SSP5-8.5 | CNRM-ESM2-1 | Standard | Medium | No |

**Table A2:** Description of ISMIP6 Antarctica Tier 3 simulations, which are performed with ocean only (OO) or atmosphere only (AO) forcing, as indicated in the CMIP AOGCM column.

| Experiment ID | Scenario | CMIP AOGCM | Standard/Open | Ocean forcing | Fracture |
|---|---|---|---|---|---|

| | | | | | |
|---|---|---|---|---|---|
| expC1 | RCP8.5 | NorESM1-M AO | N/A | Medium | No |
| expC2 | RCP8.5 | NorESM1-M OO | Open | Medium | No |
| expC3 | RCP8.5 | NorESM1-M OO | Standard | Medium | No |
| expC4 | RCP8.5 | MIROC-ESM-CHEM AO | N/A | Medium | No |
| expC5 | RCP8.5 | MIROC-ESM-CHEM OO | Open | Medium | No |
| expC6 | RCP8.5 | MIROC-ESM-CHEM OO | Standard | Medium | No |
| expC7 | RCP2.6 | NorESM1-M AO | N/A | Medium | No |
| expC8 | RCP2.6 | NorESM1-M OO | Open | Medium | No |
| expC9 | RCP2.6 | NorESM1-M OO | Standard | Medium | No |
| expC10 | RCP8.5 | CCSM4 AO | N/A | Medium | No |
| expC11 | RCP8.5 | CCSM4 OO | Open | Medium | No |
| expC12 | RCP8.5 | CCSM4 OO | Standard | Medium | No |

**Table A3:** Description of ISMIP6 Antarctica Tier 3 simulations to investigate impact of ocean forcing.

| Experiment ID | Scenario | CMIP AOGCM | Standard/Open | Ocean forcing | Fracture |
|---|---|---|---|---|---|
| expD1 | RCP8.5 | MIROC-ESM-CHEM | Standard | High | No |
| expD2 | RCP8.5 | MIROC-ESM-CHEM | Standard | Low | No |
| expD3 | RCP2.6 | NorESM1-M | Standard | High | No |
| expD4 | RCP2.6 | NorESM1-M | Standard | Low | No |
| expD5 | RCP8.5 | CCSM4 | Standard | High | No |
| expD6 | RCP8.5 | CCSM4 | Standard | Low | No |
| expD7 | RCP8.5 | HadGEM2-ES | Standard | High | No |
| expD8 | RCP8.5 | HadGEM2-ES | Standard | Low | No |
| expD9 | RCP8.5 | CSIRO-Mk3.6.0 | Standard | High | No |
| expD10 | RCP8.5 | CSIRO-Mk3.6.0 | Standard | Low | No |
| expD11 | RCP8.5 | IPSL-CM5A-MR | Standard | High | No |
| expD12 | RCP8.5 | IPSL-CM5A-MR | Standard | Low | No |
| expD13 | SSP5-8.5 | CNRM-CM6-1 | Standard | High | No |
| expD14 | SSP5-8.5 | CNRM-CM6-1 | Standard | Low | No |
| expD15 | SSP5-8.5 | UKESM1-0-LL | Standard | High | No |
| expD16 | SSP5-8.5 | UKESM1-0-LL | Standard | Low | No |
| expD17 | SSP5-8.5 | CESM2 | Standard | High | No |
| expD18 | SSP5-8.5 | CESM2 | Standard | Low | No |
| … | … | … | … | … | … |
| expD51 | RCP8.5 | NorESM1-M | Standard | PIGL gamma calibration Low | No |
| expD52 | RCP8.5 | NorESM1-M | Standard | PIGL gamma calibration High | No |
| expD53 | RCP8.5 | MIROC-ESM-CHEM | Standard | PIGL gamma calibration Medium | No |
| expD54 | RCP8.5 | MIROC-ESM-CHEM | Standard | PIGL gamma calibration Low | No |
| expD55 | RCP8.5 | MIROC-ESM-CHEM | Standard | PIGL gamma calibration High | No |
| expD56 | RCP8.5 | CCSM4 | Standard | PIGL gamma calibration Medium | No |
| expD57 | RCP8.5 | CCSM4 | Standard | PIGL gamma calibration Low | No |

| expD58 | RCP8.5 | CCSM4 | Standard | PIGL gamma calibration High | No |

**Table A4:** Description of ISMIP6 Antarctica Tier 3 simulations to investigate impact of ice shelf fracture.

| Experiment ID | Scenario | CMIP AOGCM | Standard/Open | Ocean forcing | Fracture |
|---|---|---|---|---|---|
| expE1 | RCP8.5 | NorESM1-M | Open | Medium | Yes |
| expE2 | RCP8.5 | MIROC-ESM-CHEM | Open | Medium | Yes |
| expE3 | RCP8.5 | HadGEM2-ES | Open | Medium | Yes |
| expE4 | RCP8.5 | CSIRO-Mk3.6.0 | Open | Medium | Yes |
| expE5 | RCP8.5 | IPSL-CM5A-MR | Open | Medium | Yes |
| expE6 | RCP8.5 | NorESM1-M | Standard | Medium | Yes |
| expE7 | RCP8.5 | MIROC-ESM-CHEM | Standard | Medium | Yes |
| expE8 | RCP8.5 | HadGEM2-ES | Standard | Medium | Yes |
| expE9 | RCP8.5 | CSIRO-Mk3.6.0 | Standard | Medium | Yes |
| expE10 | RCP8.5 | IPSL-CM5A-MR | Standard | Medium | Yes |
| expE11 | SSP5-8.5 | CNRM-CM6-1 | Open | Medium | Yes |
| expE12 | SSP5-8.5 | UKESM1-0-LL | Open | Medium | Yes |
| expE13 | SSP5-8.5 | CESM2 | Open | Medium | Yes |
| expE14 | SSP5-8.5 | CNRM-ESM2-1 | Open | Medium | Yes |
| expE15 | SSP5-8.5 | CNRM-CM6-1 | Standard | Medium | Yes |
| expE16 | SSP5-8.5 | UKESM1-0-LL | Standard | Medium | Yes |
| expE17 | SSP5-8.5 | CESM2 | Standard | Medium | Yes |
| expE18 | SSP5-8.5 | CNRM-ESM2-1 | Standard | Medium | Yes |

**Table A5:** Description of ISMIP6 Greenland Tier 2 simulations

| Experiment ID | Scenario | CMIP AOGCM | Ocean forcing |
|---|---|---|---|
| expa01 | RCP8.5 | IPSL-CM5A-MR | Medium |
| expa02 | RCP8.5 | CSIRO-Mk3.6.0 | Medium |
| expa03 | RCP8.5 | ACCESS1.3 | Medium |
| expb01 | SSP5-8.5 | CNRM-CM6-1 | Medium |
| expb02 | SSP1-2.6 | CNRM-CM6-1 | Medium |
| expb03 | SSP5-8.5 | UKESM1-0-LL | Medium |
| expb04 | SSP5-8.5 | CESM2 | Medium |
| expb05 | SSP5-8.5 | CNRM-ESM2-1 | Medium |

**Table A6:** Description of ISMIP6 Greenland Tier 3 simulations, which are performed with ocean only (OO) or atmosphere only (AO) forcing, as indicated in the CMIP AOGCM column.

| Experiment ID | Scenario | CMIP AOGCM | Ocean forcing |
|---|---|---|---|
| expc01 | RCP8.5 | MIROC5 AO | Medium |
| expc02 | RCP8.5 | MIROC5 OO | Medium |
| expc03 | RCP8.5 | CSIRO-Mk3.6.0 AO | Medium |
| expc04 | RCP8.5 | CSIRO-Mk3.6.0 OO | Medium |
| expc05 | RCP2.6 | MIROC5 AO | Medium |
| expc06 | RCP2.6 | MIROC5 OO | Medium |
| expc07 | RCP8.5 | NorESM1-M AO | Medium |
| expc08 | RCP8.5 | NorESM1-M OO | Medium |

| | | | |
|---|---|---|---|
| expc09 | RCP8.5 | MIROC5 OO | High |
| expc10 | RCP8.5 | MIROC5 OO | Low |

**Table A7:** Description of ISMIP6 Greenland Tier 3 simulations to investigate impact of ocean forcing.

| Experiment ID | Scenario | CMIP AOGCM | Ocean forcing |
|---|---|---|---|
| expd01 | RCP8.5 | NorESM1-M | High |
| expd02 | RCP8.5 | NorESM1-M | Low |
| expd03 | RCP8.5 | HadGEM2-ES | High |
| expd04 | RCP8.5 | HadGEM2-ES | Low |
| expd05 | RCP2.6 | MIROC5 | High |
| expd06 | RCP2.6 | MIROC5 | Low |
| expd07 | RCP8.5 | IPSL-CM5A-MR | High |
| expd08 | RCP8.5 | IPSL-CM5A-MR | Low |
| expd09 | RCP8.5 | CSIRO-Mk3.6.0 | High |
| expd10 | RCP8.5 | CSIRO-Mk3.6.0 | Low |
| expd11 | RCP8.5 | ACCESS1.3 | High |
| expd12 | RCP8.5 | ACCESS1.3 | Low |
| expd13 | SSP5-8.5 | CNRM-CM6-1 | High |
| expd14 | SSP5-8.5 | CNRM-CM6-1 | Low |
| expd15 | SSP1-2.6 | CNRM-CM6-1 | High |
| expd16 | SSP1-2.6 | CNRM-CM6-1 | Low |
| expd17 | SSP5-8.5 | UKESM1-0-LL | High |
| expd18 | SSP5-8.5 | UKESM1-0-LL | Low |
| expd19 | SSP5-8.5 | CESM2 | High |
| expd20 | SSP5-8.5 | CESM2 | Low |
| expd21 | SSP5-8.5 | CNRM-ESM2-1 | High |
| expd22 | SSP5-8.5 | CNRM-ESM2-1 | Low |

**Appendix B: ISMIP6 grids and variable request**

To facilitate intercomparison of model submissions, groups are requested to submit on the ISMIP6 Antarctica and Greenland regular grids at a resolution that is the closest to the modeled ice sheet grid. For Antarctica, the ISMIP6 the grid is a polar stereo-graphic projection, with standard parallel at 71° S and central meridian of 0° W on datum WGS84. The lowest left corner is at x = -3040 km and y = -3040 km, while the upper right corner is at x = 3040 km and y = 3040 km. Acceptable resolutions are 32 km, 16 km, 8 km, 4 km, 2 km or 1 km. Submissions will be stored on the submitted resolution for archiving and conservatively interpolated by ISMIP6 to the 8 km Antarctica grid for intercomparison. For Greenland, the ISMIP6 grid is a polar stereo-graphic projection, with standard parallel at 70° N and central meridian of 45° W on datum WGS84. The lowest left corner is at x = -720 km and y = -3450 km, while the upper right corner is at x = 960 km and y = -570 km. Acceptable resolutions are 20 km, 10 km, 5 km, 4km, 2 km or 1 km. Submissions will be stored on the submitted resolution for archiving and conservatively interpolated by ISMIP6 to the 5 km Greenland grid for intercomparison.

The ice sheet data request (Table B1) contains key characteristics needed to evaluate the ice sheet geometry, and ice sheet flow. It also contains key ice sheet specific boundary conditions that may differ between models and a record of the forcing applied to the ice sheet model. To facilitate the analysis of the ice sheet contribution to sea level, a number of integrated measures (for example, ice sheet mass) are also requested. Two dimensional state variables (ST) are requested as yearly

snapshot corresponding to the end of the year in a simulation for state variables (such as ice thickness), and as yearly average for flux variables (FL, such as surface mass balance). Fields such as surface mass balance flux should be what was applied as a boundary condition to the ice sheet model and may be different from the input forcing file.

5 **Table B1:** Data request for the dynamical ice sheet model submissions. These fields, if applicable to the model, are saved on the regular ISMIP6 ice sheet grid that is the closest to a model native grid and contain yearly output. Type indicates whether the variable is a state variable (ST) or a flux variable (SF).

| Long name (netCDF) | Units | Standard Name (CF) | Type |
|---|---|---|---|
| Two dimensional variables | | | |
| Ice Sheet Altitude | m | surface_altitude | ST |
| Ice Sheet Thickness | m | land_ice_thickness | ST |
| Bedrock Altitude | m | bedrock_altitude | ST |
| Base Elevation | m | base_altitude | ST |
| Land ice thickness imbalance | m s$^{-1}$ | tendency_of_land_ice_thickness | ST |
| Bedrock Geothermal Heat Flux | W m$^{-2}$ | upward_geothermal_heat_flux_at_ground_level | FL |
| Land ice calving flux | kg m$^{-2}$ s$^{-1}$ | land_ice_specific_mass_flux_due_to_calving | FL |
| Land ice vertical front mass balance flux | kg m$^{-2}$ s$^{-1}$ | land_ice_specific_mass_flux_due_to_calving_and_ice_front_melting | FL |
| Grounding line flux | kg m$^{-2}$ s$^{-1}$ | land_ice_specific_mass_flux_due_at_grounding_line | FL |
| Surface Mass Balance flux | kg m$^{-2}$ s$^{-1}$ | land_ice_surface_specific_mass_balance_flux | FL |
| Basal Mass Balance of grounded ice sheet | kg m$^{-2}$ s$^{-1}$ | land_ice_basal_specific_mass_balance_flux | FL |
| Basal Mass Balance of floating ice shelf | kg m$^{-2}$ s$^{-1}$ | land_ice_basal_specific_mass_balance_flux | FL |
| X-component of land ice surface velocity | m s$^{-1}$ | land_ice_surface_x_velocity | ST |
| Y-component of land ice surface velocity | m s$^{-1}$ | land_ice_ surface_y_velocity | ST |
| Z-component of land ice surface velocity | m s$^{-1}$ | land_ice_ surface_upward_velocity | ST |
| X-component of land ice basal velocity | m s$^{-1}$ | land_ice_basal_x_velocity | ST |
| Y-component of land ice basal velocity | m s$^{--1}$ | land_ice_basal_y_velocity | ST |
| Z-component of land ice basal velocity | m s$^{-1}$ | land_ice_basal_upward_velocity | ST |
| X-component of land ice vertical mean velocity | m s$^{-1}$ | land_ice_vertical_mean_x_velocity | ST |
| Y-component of land ice vertical mean velocity | m s$^{-1}$ | land_ice_vertical_mean_y_velocity | ST |
| Land ice basal drag | Pa | land_ice_basal_drag | ST |
| Surface Temperature | K | temperature_at_top_of_ice_sheet_model | ST |
| Basal Temperature of Grounded Ice Sheet | K | temperature_at_base_of_ice_sheet_model | ST |
| Basal Temperature of Floating Ice Shelf | K | temperature_at_base_of_ice_sheet_model | ST |

| | | | |
|---|---|---|---|
| Land ice area fraction | % | land_ice_area_fraction | ST |
| Grounded ice area fraction | % | grounded_ice_sheet_area_fraction | ST |
| Floating ice shelf area fraction | % | floating_ice_shelf_area_fraction | ST |

| Scalar outputs / Integrated measures | | | |
|---|---|---|---|
| Ice Mass | kg | land_ice_mass | ST |
| Ice Mass not displacing sea water | kg | land_ice_mass_not_displacing_sea_water | ST |
| Area covered by grounded ice | $m^2$ | grounded_ice_sheet_area_ | ST |
| Area covered by floating ice | $m^2$ | floating_ice_shelf_area | ST |
| Total SMB flux | kg s$^{-1}$ | tendency_of_land_ice_mass_due_to_surface_mass_balance | FL |
| Total BMB flux | kg s$^{-1}$ | tendency_of_land_ice_mass_due_to_basal_mass_balance | FL |
| Total calving flux | kg s$^{-1}$ | tendency_of_land_ice_mass_due_to_calving | FL |
| Total calving and ice front melting flux | kg s$^{-1}$ | tendency_of_land_ice_mass_due_to_calving_and_ice_front_melting | FL |
| Total grounding line flux | kg s$^{-1}$ | tendency_of_grounded_ice_mass | FL |

## Appendix C: Antarctic atmospheric forcing preparation

In general, all files were obtained from the CMIP distribution through the Earth System Grid Federation (ESGF). As indicated in Table C1, the first ensemble member for each model and CMIP experiment was selected. (The ensemble member is denoted in CMIP as "r1i1p1", where "r" is a center-designated realization number, "i" is the initialization number, and "p" is the physics number. For CMIP6, a center-designated ensemble forcing number "f" is also used, but this mostly corresponds to the CMIP experiment.). The CESM2 datasets originated from the initial "MOAR" run, and provided directly to ISMIP6 prior to this dataset becoming available on the ESGF grid.

The primary atmospheric variables to be used are precipitation (pr), evaporation (evspsbl), runoff (mrro, mrros), and skin temperature (ts). As defined by the ISMIP6 protocol (Nowicki et al., 2016), surface mass balance (SMB) is the net of precipitation minus evaporation minus runoff. In general, these were taken from the ESGF atmosphere "Realm" of variables, and are defined globally so as to accommodate an ISM grid extending beyond continental boundaries. In the CMIP5 ESM output, the runoff variable from the land surface Realm is often problematic over ice sheets. There are two runoff variables – "surface" and "total" – which are ambiguously defined. In many cases one or both of the variables incorporates a restoration to the ocean of the accumulated SMB for maintaining mass equilibrium in the absence of a dynamical ice sheet model. This is commonly referred to as a Poor Man's iceberg calving, or "frozen runoff". Liquid runoff is currently a negligible term for continental-averaged Antarctic SMB but could conceivably become locally significant by 2100 on ice shelves and in coastal regions, particularly along the northern Antarctic Peninsula.  Hence reasonable effort was made to incorporate a runoff variable into the forcing from the CMIP simulations where available.

The CMIP experiment output that was used are from the "historical" experiment, and from the 21st Century Representative Concentration Pathways (RCPs) 2.6 and 8.5 from CMIP5, and Shared Socioeconomic Pathways (SSPs) 126

and 585 of CMIP6. A climatology of each variable is first constructed corresponding to the annual average over model years 1995-2014. In CMIP5, the historical experiment ends at model year 2004, so that the climatology is produced from both the historical and RCP simulation output. In CMIP6, the historical experiment ends at model year 2014, and the climatology is taken entirely from the historical simulation output. Using the climatology, annual anomalies are then computed for each

variable over the period 1950-2100. The climatology and the anomalies are then re-gridded to an azimuthal equal-area grid designated for ISMIP6 with 8 km spacing. In general, the CMIP ESM native grid spacing at high latitudes is very fine zonally and relatively coarse meridionally, and this produces artifacts when applying conservative interpolation methods. Here we have used cubic spline interpolation in the transfer from native resolution to the 8-km grid, and conservative interpolation from the 8-km grid to other ISMIP6 grids. The CMIP ESM models selected have a range in native latitudinal grid spacing from less

that 1 degree to more that 2.8 degrees.

        The SMB datasets are in units of kg m$^{-2}$ s$^{-1}$ water equivalent and need to be converted by users to m yr$^{-1}$ ice equivalent via:

$$aSMB[\text{m yr}^{-1}] = aSMB[\text{kg m}^{-2}\text{s}^{-1}] \times 31556926\,[\text{s yr}^{-1}] \times \left(\frac{1}{1000}\right)[\text{m}^3\,\text{kg}^{-1}] \times \left(\frac{\rho_w}{\rho_i}\right)$$

where $\rho_w$ and $\rho_i$ are the densities of water and ice (typically, 1000.0 kg m$^{-3}$ and 917.0 kg m$^{-3}$), respectively. The temperature

datasets are provided in units of degrees Kelvin.

**Table C1:** CMIP models used to create surface mass balance and surface temperature forcing for ice sheet models.

| Model | Scenario | Ensemble Member | Runoff Variable Used | Native Grid | Native Pole Point | Notes |
|---|---|---|---|---|---|---|
| CCSM4 | hist/rcp85 | r1i1p1 | mrro | 192 × 288 (0.94° × 1.25°) | No | ▪ mrro has overlapping months for 2005 in original historical and rcp scenario files. |
| CCSM4 | hist/rcp26 | r1i1p1 | mrro | 192 × 288 (0.94° × 1.25°) | No | |
| MIROC-ESM-CHEM | hist/rcp85 | r1i1p1 | mrro | 64 × 128 (2.81° × 2.81°) | No | |
| MIROC-ESM-CHEM | hist/rcp26 | r1i1p1 | mrro | 64 × 128 (2.81° × 2.81°) | No | |
| NorESM1-M | hist/rcp85 | r1i1p1 | mrro* | 94 × 144 (1.91° × 2.50°) | Yes | ▪ *Runoff computed from daily files. |
| NorESM1-M | hist/rcp26 | r1i1p1 | mrro* | 94 × 144 (1.91° × 2.50°) | Yes | |
| HadGEM2-ES | hist/rcp85 | r1i1p1 | mrros* | 145 × 192 (1.25° × 1.88°) | Yes | ▪ *Runoff supplied by Robin. |
| CSIRO-Mk3.6.0 | hist/rcp85 | r1i1p1 | N/A | 96 × 192 (1.88° × 1.88°) | No | No viable runoff. |
| IPSL-CM5A-MR | hist/rcp85 | r1i1p1 | N/A | 143 × 144 (1.27° × 2.50°) | Yes | No viable runoff. |
| UKESM1-0-LL | hist/ssp585 | r1i1p1f2 | mrro | 144 × 192 (1.25° × 1.88°) | No | ▪ Evaporation computed from latent heat flux (hfls). |

| | | | | | | |
|---|---|---|---|---|---|---|
| CNRM-CM6-1 | hist/ssp126 | r1i1p1f2 | mrros | 128 × 256 (1.41° × 1.41°) | No | |
| CNRM-CM6-1 | hist/ssp585 | r1i1p1f2 | mrros | 128 × 256 (1.41° × 1.41°) | No | |
| CNRM-ESM2-1 | hist/ssp585 | r1i1p1f2 | mrros | 128 × 256 (1.41° × 1.41°) | No | |
| CESM2 | hist/ssp585 | "MOAR" run | QRUNOFF_ICE | 192 × 288 (0.94° × 1.25°) | Yes | ▪ All variables supplied by Kate Thayer-Calder, Bill Lipscomb. ▪ Evaporation undefined over ocean. |

**Appendix D: Implementation of Greenland tidewater glacier retreat parameterization in ice sheet models**

We describe the method and implementation of tidewater glacier forcing used in large-scale ice sheet models with possibly relatively coarse resolution and with initial geometries that can differ from observations. The approach we are proposing is
5 implementing a time-dependent set of retreat masks, that define the maximum calving front position at any time during an experiment. Differences in initial ice sheet model geometry requires that the retreat masks are calculated specifically for individual ice sheet models. The procedure is a mapping operation to translate the retreat, originally derived for the observed ice sheet, to the individual model geometry. Furthermore, coarse resolution models have to consider a form of sub-grid implementation to reduce biases when the calving front retreats across grid cells of large horizontal extent. We assume in the
10 following that the time dependent ice sheet retreat around Greenland is known for groups of marine-terminating outlet glaciers in seven different regions, as described by Slater et al. (2019a).

**D1 Retreat masks**

Here we first discuss the hypothetical case where an ice sheet model of very high spatial resolution has been initialised with ice front/ grounding line positions in perfect agreement with observations. We will assume that the model grid (MG) is identical
to a regular observational grid (OG), where the ice sheet geometry is defined (e.g. Morlighem et al., 2017). To determine the retreat masks we apply the following procedure:

1.  Identify bed below sea-level and in connection with the ocean on OG (Fig. D1).
    a.  define mask of ice grounded below sea-level
    b.  search all connected grid points starting at the marine margin
2.  Identify shortest distance from the ice front/grounding line for all points on OG identified in 1 along sections of bed below sea-level (Fig. D1).
    a.  use mask of connected points as defined in 1
    b.  define ocean mask
    c.  calculate distance to the nearest ocean point (2b) for all points in (2a)
25 3.  Define retreat masks by thresholding the distance map (2) for given retreat distances per region (not shown).

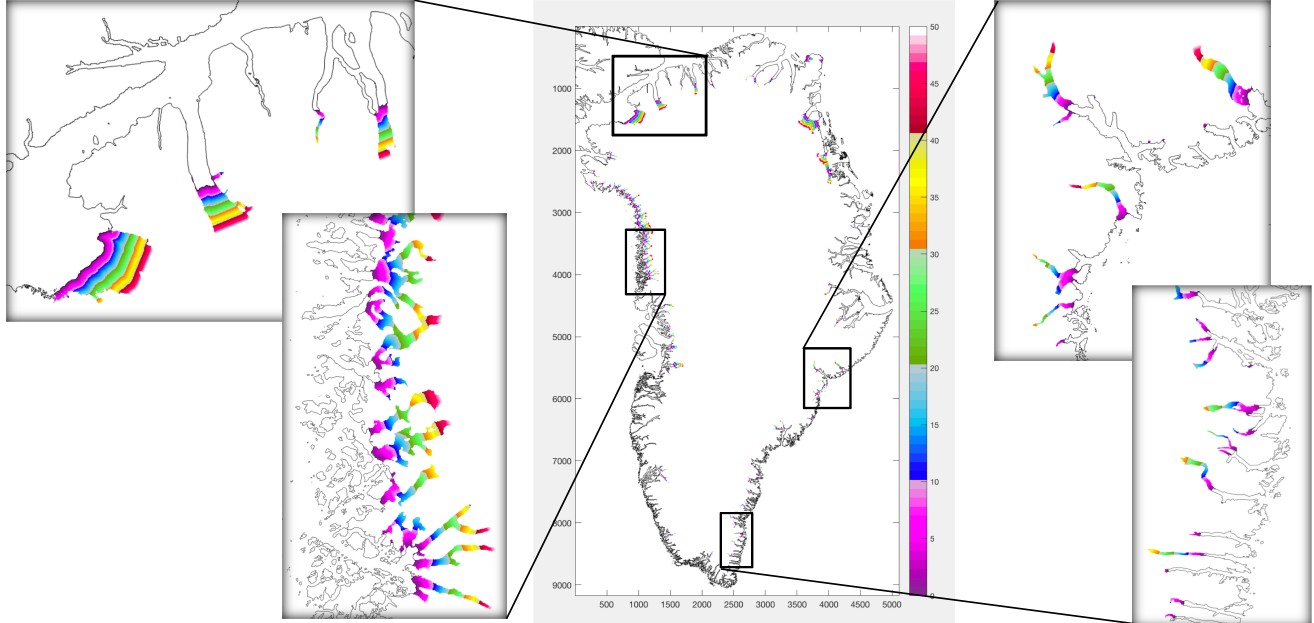

**Figure D1.** Distances from the ocean for all points identified as ice grounded on bed below sea-level in potential contact with the ocean. The data are masked to areas less than 50 km from the nearest grounding line.

**D2 Procedure for specific models**

Because the modelled initial ice sheet mask is generally different from the observed, additional complications arise because the modelled and observed glacier fronts cannot be assumed to be closely corresponding. In the model the glacier fronts may lie further out or in and different glaciers may fall together. Furthermore, the model may not resolve individual outlet glaciers due to limited resolution of a coarse grid. We therefore perform the distance calculations on OG and determine connectivity according to the observed geometry, but based on the *modelled* ice mask. The procedure described in Section D1 is augmented with interpolation steps between MG and OG.

0.  Find the modelled ice front positions and interpolate to OG.
    a.  define the mask of grounded ice on MG (threshold area fractions to get a binary mask)
    b.  interpolate grounded ice mask to OG using binned regridding (see above).
1.  Identify bed below sea-level and in connection with the ocean on OG for the modelled ice mask.
    a.  use mask of ice grounded below sea-level from 0
    b.  find all connected grid points starting at the modelled marine margin
2.  Identify shortest distance from the ice front/grounding line for all points on OG identified in 1 along sections of bed below sea-level.
    a.  use mask of connected points as defined in 1 a,b
    b.  define ocean mask
    c.  calculate distance to the nearest ocean point (2b) for all points in (2a)
3.  Remap distances found in 2c from OG to MG.
    a.  use binned regridding with masked distances from 2
4.  Identify grid points on MG intersected by the OG grid points in 1 and determine weights as area covered by OG points on MG.
    a.  use binned regridding with mask from 1
    b.  an additional weights calculation may be needed if mask in 0 contained partial cells

Because of a general mismatch (in resolution) between OG and MG, with MG typically coarser, we translate retreat on OG to partial thinning on grid MG according to the covered area. This sub-grid process is discussed in more in detail next.

**D3 Sub-grid implementation**

The implementation of outlet glacier retreat in a coarse grid model requires a form of sub-grid process to take into account partial retreat. This is needed because the two end members (1- retreat only full grid cells that are entirely ice free, 2 - retreat full grid cells already when becoming partially ice free) are under- and overestimating the retreat, respectively. This problem is illustrated in Figure D2a. The grey shading shows grid cells on the high resolution OG that fall within the footprint of the coarser MG (orange shading).

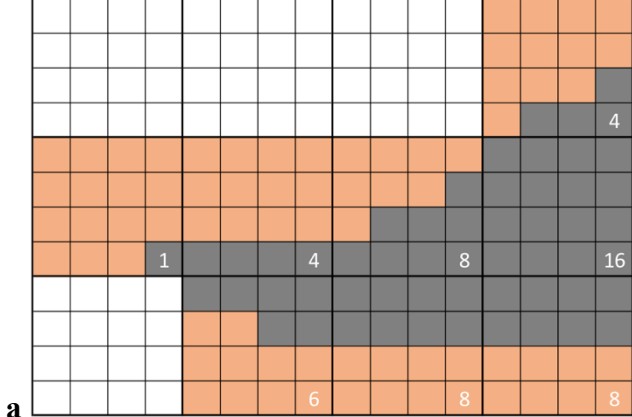 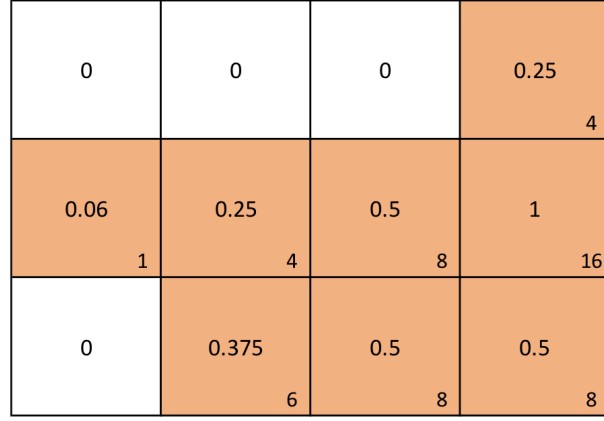

**Figure D2.** Schematic of the sub-grid implementation to translate the retreat mask defined on a high resolution observational grid (a) to the coarser resolution (b) of an ice sheet model.

The method we have tested is to translate partial retreat to partial thinning, relative to a reference thickness applied once a year. This gave comparable results for a test case of different grid resolutions from 5-20 km resolution (not shown). The limitation to apply the relative thinning once a year is required to avoid time step dependence for different models. The thinning relative to a reference thickness avoids a non-linear thinning with time. Applying partial thinning without a reference thickness (once a year) has been shown to overestimate retreat, because the thickness is exponentially decreasing and approaching the upper end member of full retreat.

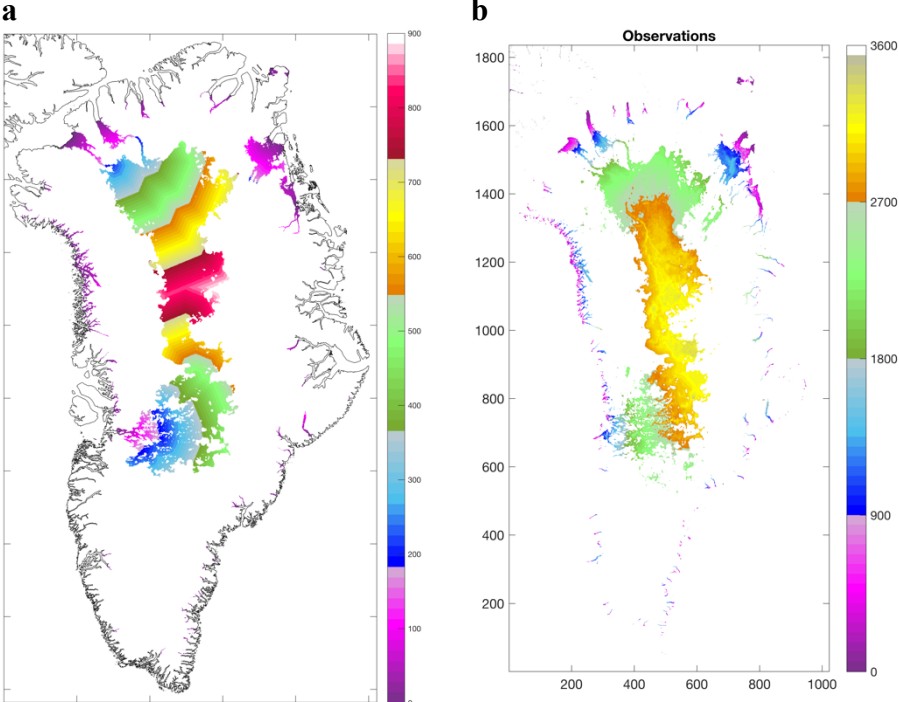

**Figure D3.** a) Distance from the ocean for all ice sheet points identified as in potential contact with the ocean. b) Ice thickness of all points connected to the ocean.

## D4 Discussion

The method has been developed for tidewater glaciers, which are the predominant form of marine-termination around Greenland. The few glaciers with floating ice tongues are not treated differently. Extending the framework for floating ice shelves would require to prescribe grounding line positions, which is not possible in the present ice sheet models. For the few outlet glaciers with floating ice at the termini, the ice sheet response will likely be underestimated, because removing floating may be expected to be less effective to speed up glaciers upstream compared to directly removing ice at a calving front. Compared to the other uncertainties associated with this method we considered this a minor effect. In our application, most of the outlet glaciers are expected to retreat in the future. However, the method allows for re-advance of glaciers up to the initial ice mask. Nevertheless, re-advance can only happen by the ice flow into formerly vacated grid cells, as the method does not 'create' mass. A partial retreat mechanism has to be considered to avoid over- or under-estimation of the retreat, in particular in coarse resolution models. This sub-grid process is implemented as partial thinning.

## Appendix E: Acronym List

AOGCM: atmosphere-ocean general circulation model

CliC: Climate and Cryosphere

CMIP5 or CMIP6: Coupled Model Intercomparison Project – Phase 5 or Phase 6

ESGF: Earth System Grid Federation

ESM: Earth-system models

FL: Flux variable

IMBIE2: Ice sheet Mass Balance Inter-comparison Exercise

IPCC: Intergovernmental Panel on Climate Change

ISMIP6: Ice Sheet Model Intercomparison Project for CMIP6

ISM: Ice-sheet models

ISM-ESM: ice-sheet models are fully coupled within Earth-system models

10 MAR: Modèle Atmosphérique Régionale

MeanAnt: Calibrations based on observed mean sub-shelf basal melt over Antarctica

MEOP: Marine Mammals Exploring the Oceans from Pole to Pole

MG: Model grid

OG: Observational grid

20 PIGL: Pine Island Grounding Line calibration

RACMO2.3p2: Regional Atmospheric Climate Model version 2.3p2

RCM: Regional climate model

RCP: Representative Concentration Pathways

SMB: Surface mass balance

30 SSP: Shared Socioeconomic Pathways

ST: State variable

WCRP: World Climate Research Programme

WGS84: World Geodetic System 1984

WOA18: World Ocean Atlas 2018

40 *Data availability.* All of the projection datasets described in this paper are freely available from the ISMIP6 ftp server hosted at the University at Buffalo; access can be obtained by emailing ismip6@gmail.com. CMIP5 model output is available at https://esgf-node.llnl.gov/projects/esgf-llnl/ (last access April 2019). CMIP6 model output is available at https://esgf-node.llnl.gov/search/cmip6/ (last access September 2019). The MAR based Greenland projections are available on ftp://ftp.climato.be/fettweis/MARv3.9/ISMIP6/GrIS/ (last access April 2019). The CESM2 MOAR data sets became the initial 45 CESM2(CAM6) future scenario simulations submitted to the CMIP6 archive, which were then retracted in April 2020 because both anthropogenic and biomass burning secondary organic aerosol emissions were set to zero starting in (model date) 2015 in error. These data sets were replaced by the corrected ones in May 2020 on ESGF. Many aspects of the simulation characteristics between the erroneous and corrected experiments are very similar, with differences within the limits of internal variability. Therefore, most results and conclusions based on the previous simulations remain valid, but the results will differ 50 in detail and in their internal variability.

*Competing interests.* Xavier Fettweis is a member of the editorial board of the journal. Sophie Nowicki, Ayako Abe-Ouchi, Helene Seroussi, Robin Smith and Bill Lipscomb are editors of the ISMIP6 special issue of The Cryosphere.

55 *Acknowledgements.* We thank the Climate and Cryosphere (CliC) effort, which provided support for ISMIP6 through sponsoring of workshops, hosting the ISMIP6 website and wiki, and promoted ISMIP6. We acknowledge the World Climate

Research Programme, which, through its Working Group on Coupled Modelling, coordinated and promoted CMIP5 and CMIP6. We thank the climate modeling groups for producing and making available their model output, the Earth System Grid Federation (ESGF) for archiving the CMIP data and providing access, the University at Buffalo for ISMIP6 data distribution and upload, and the multiple funding agencies who support CMIP5 and CMIP6 and ESGF. We thank the ISMIP6 steering committee, the ISMIP6 model selection group and ISMIP6 dataset preparation group for their continuous engagement in defining ISMIP6. Sophie Nowicki, Helene Seroussi, Richard Cullather, Eric Larour, Isabel Nias and Erika Simon were supported by grants from the NASA Cryospheric Sciences, Sea Level Change Team and Modeling, Analysis and Predictions Programs. Denis Felikson was supported by an appointment to the NASA Postdoctoral Program at the Goddard Space Flight Center, administered by Universities Space Research Association under contract with NASA. Heiko Goelzer, Peter Kuipers Munneke, Roderik van de Wal and Michiel van den Broeke acknowledge support from the Netherlands Earth System Science Centre (NESSC). Support for Xylar Asay-Davis was provided through the Scientific Discovery through Advanced Computing (SciDAC) program funded by the US Department of Energy (DOE), Office of Science, Advanced Scientific Computing Research and Biological and Environmental Research Programs. Tore Hattermann was supported by a Norwegian Research Council grant 280727. Fiamma Straneo and Donald Slater acknowledge support from NSF 1916566 and NASA NNX17AI03G. Thomas Bracegirdle was supported both by the UK Natural Environment Research Council through the British Antarctic Survey research programme Polar Science for Planet Earth and the Scientific Committee on Antarctic Research (SCAR) AntClim21 Research Programme. Jonathan Gregory and Robin S. Smith were supported by NCAS, funded by the UK National Environment Research Council. Alice Barthel was supported by the U.S. Department of Energy (DOE) Early Career Program and Biological and Environmental Research Program (HiLAT project). This material is based in part upon work supported by the National Center for Atmospheric Research, which is a major facility sponsored by the National Science Foundation under Cooperative Agreement No. 1852977. This is ISMIP6 contribution No 9.

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
