# Peer review of "Experimental protocol for sea level projections from ISMIP6 standalone ice sheet models"

_The Cryosphere, 2019_

## Referee Comment (RC1) · Anonymous Referee #1 · 10 Mar 2020

**Review: "Experimental protocol for sea level projections from ISMIP6 standalone ice sheet models"**

by Nowicki et al.

Submitted to *The Cryosphere*

**1  General**

In this paper, the authors describe the framework for the ISMIP6 numerical experiments. This is clearly an important piece of work that documents a tremendous amount of effort, minutia, and thought. At the outset, it was not clear to me that a peer-reviewed scientific publication is the best venue for such a manuscript. However, given the readership of the *The Cryosphere*, I think that this is a fair choice and would support publication after a few minor revisions.

**2  Remarks**

1. From my perspective, the compilation of surface temperature anomalies and surface mass balance anomalies for several different models for both Greenland and Antarctica is very interesting. These are plots that I will likely refer back to and possibly use in talks. That said, I am confused about whether these results are published elsewhere and included here for succinctness or if this is their first presentation. If, indeed, this is their first presentation, I suggest highlighting this fact in the paper to a greater extent.

2. The number of acronyms in this paper is off the charts. I understand that this comes with the territory, yet it is still a hurdle to understanding the contents of this paper. I suggest (a) a table of acronyms in the appendix before the list of tier 2 simulations and (b) at every instance possible, avoid using an acronym or use both words and acronyms. I would understand if the authors find this request difficult to implement, my main request is that they think critically about whether or not every acronym is actually required and make an effort to reduce the total number.

3. Numerous 'under review' papers are cited. This makes sense because this paper and the cited papers are pieces of a larger puzzle, however, it would be ideal for the authors of this paper to explain the results of the cited papers to a greater degree, given that the referees have no access to the contents of those papers. In the future, this will also be beneficial as it will highlight the connections between each piece of the ISMIP6 puzzle.

**3  Specific comments**

1. page 11, line 23: here the 'ISM' is in the subscript whereas in other places, e.g. line 25, it is not. Which one is correct? I find the double subscript cumbersome but also think that the $T\_ISM_{RCM}$ notation is difficult to wrap my head around.

2. page 15, line 4: here and elsewhere, e.g. equation (7), I suggest removing the $\times$ symbol.

3. page 17, line 15: what role does sliding due to subglacial hydrology play in these experiments? Here subglacial discharge primarily affects melting at the front, yet could also substantially affect sliding, which would be worth mentioning.

4. page 18, line 21: I find the presentation of this conditional statement a bit odd. It is possible that this is due to the weird spacing, but my main thought is that it is not clear what the intent of the presentation. Possibly a table or flowchart describing the different retreat scenarios would be better?

5. figure 9 and 10: these figures are extremely small and are a little difficult to read for that reason.

6. page 20, section 6: how does the ongoing discussion of the 'marine ice cliff instability' play into these choices?

---

## Referee Comment (RC2) · Anonymous Referee #2 · 12 Mar 2020

General comments

This paper documents the experimental setup for standalone ice sheet model inter-comparison experiments using a variety of climate model forcings from CMIP6. This is a tremendous undertaking and these kinds of experiments are vital in assessing the sources of uncertainty in projections of sea-level rise over the coming century. While I recommend the paper for publication with a few revisions, I have some concerns about the design of the experiments and what they aim to test. At the same time, I realize that this project has to meet the conflicting demands of running a thorough experiment that tests a wide range of parameters, and creating a protocol that ice sheet modelers can follow.

Specific comments

[Figure]

The experiment aims to test the sensitivity of sea-level projects to many variables: which ice sheet model is used, which future anthropogenic forcing scenario (RCP2.6 vs 8.5) is used, which atmosphere-ocean general circulation model is used, and how the outputs of each GCM are downscaled from their native resolution to the relatively finer resolution of ice sheet models. The authors themselves state that the AOGCM forcing needs to be supplied to ice sheet models in a "uniform, standardized manner" (pg. 2 lines 29-30). While this protocol will serve as a valuable guide for future sea-level projection experiments, I think there are two respects in which the uniformity could be improved.

The experimental design uses six AOGCMs for Greenland and and six for Antarctica, but only two models are common to each (CSIRO-Mk3.6 and HadGEM2-ES). Several others are clearly related, for example, MIROC5 and MIROC-ESM-CHEM, but there's nothing in the text explaining the differences. The authors cite Barthel et al. which is currently in review but some summary of the differences would be worthwhile. It's not my place to review the Barthel et al. paper, but including GCMs that meet the criteria for inclusion for only one ice sheet but not the other is a departure from the authors' stated goal of uniformity. An argument could be made here that less is more. There's a similar problem with the climate forcing scenarios – RCP2.6 with some models but not others, no intermediate climate forcing.

The biggest issue I have is with the climate model downsampling. The Greenland runs use the regional climate model MAR, while Appendix C seems to say that the Antarctic climate model output was directly interpolated onto the ice sheet model grid . The authors state that using a RCM for Antarctica was prohibitively expensive. I certainly won't argue that point but several of the coauthors of this paper have run MAR for Antarctica (Agosta et al. 2019, Estimation of the Antarctic surface mass balance using the regional climate model MAR) and doing so for this study would be a big improvement. In principle the authors could test whether the downscaling or the choice of climate model had more of an effect by also interpolating the GCM output directly

for Greenland and comparing the results. But this might not be very informative for Antarctica as the two continents have different topographic relief. The experiment is consistent in using the same parameterizations to extrapolate the oceanic variables for both ice sheets, and it would be great to see the same methodology applied to the atmosphere too.

Technical corrections

Page 5, line 5: criteria

Page 6, line 6: scalar

Page 7, line 2: Time-dependent data assimilation does a much better job about transients, see e.g. Goldberg et al. 2015 or Gillet-Chaulet 2019.

Figures 2-5: These figures are difficult to parse visually. Some way of showing the difference between the RCP2.6 and RCP8.5 scenarios would be especially helpful, either by using dashed lines for one scenario in the same plot or, better yet, putting the two on different plots entirely.

Page 18: Several paragraphs repeat information that's already in Slater et al. This paper is long as it is and a shorter summary of this would cut down on length some. Likewise the discussion of the results from Jourdain et al.

Page 21, line 8: the second period... lasts

---

## Author Comment (AC1) · 8 May 2020

**Anonymous Referee #1:**

1 General
In this paper, the authors describe the framework for the ISMIP6 numerical experiments. This is clearly an important piece of work that documents a tremendous amount of effort, minutia, and thought. At the outset, it was not clear to me that a peer-reviewed scientific publication is the best venue for such a manuscript. However, given the readership of the The Cryosphere, I think that this is a fair choice and would support publication after a few minor revisions.

We thank the reviewer for this review and all the suggestions. Indeed, designing the framework for the ISMIP6 protocol is the result of the work of a whole community, and as an endorsed MIP of CMIP6, we are required by CMIP6 to record in a peer-reviewed journal the detailed protocol for our experimental framework.

2 Remarks
1. From my perspective, the compilation of surface temperature anomalies and surface mass balance anomalies for several different models for both Greenland and Antarctica is very interesting. These are plots that I will likely refer back to and possibly use in talks. That said, I am confused about whether these results are published elsewhere and included here for succinctness or if this is their first presentation. If, indeed, this is their first presentation, I suggest highlighting this fact in the paper to a greater extent.

We thank the reviewer for this suggestion. This is indeed their first presentation in the literature (and the first time that these datasets were generated). We have modified the text slightly to highlight this in sections 1, 4.1, 4.2 and 7.

2. The number of acronyms in this paper is off the charts. I understand that this comes with the territory, yet it is still a hurdle to understanding the contents of this paper. I suggest (a) a table of acronyms in the appendix before the list of tier 2 simulations and (b) at every instance possible, avoid using an acronym or use both words and acronyms. I would understand if the authors find this request difficult to implement, my main request is that they think critically about whether or not every acronym is actually required and make an effort to reduce the total number.

We thank the reviewer for this comment. We have significantly reduced the number of acronyms used, as well as provided a list of acronyms in Appendix E.

3. Numerous `under review' papers are cited. This makes sense because this paper and the cited papers are pieces of a larger puzzle, however, it would be ideal for the authors of this paper to explain the results of the cited papers to a greater degree, given that the referees have no access to the contents of those papers. In the future, this will also be beneficial as it will highlight the connections between each piece of the ISMIP6 puzzle.

Two of the "under review" papers are now published, so we revised the manuscript. The other two manuscripts are being revised. However, we revised the manuscript as suggested, and provide a summary of these related ISMIP6 publications in order to highlights the connections between each pieces of the ISMIP6 puzzle.

3 Specific comments
page 11, line 23: here the `ISM' is in the subscript whereas in other places, e.g. line

25, it is not. Which one is correct? I find the double subscript cumbersome but also think that the $T\_ISM_{RCM}$ notation is difficult to wrap my head around.

Thank you for having spotted the mistake in equation 5. We have corrected the mistake and double checked all equations. We think that it is really important to keep track of whether a field comes from an RCM or AOGCM, and whether the field refers to a forcing applied to the ice sheet model, despite resulting in a more complicated notation. We have revised our equation using $SMB_{ISM,RCM}$ , $SMB_{ISM,AOGCM}$ , $T_{ISM,RCM}$ and $T_{ISM,AOGCM}$ instead of the original notation $SMB\_ISM_{RCM}$, $SMB\_ISM_{AOGCM}$, $T\_ISM_{RCM}$ and $T\_ISM_{AOGCM}$ .

page 15, line 4: here and elsewhere, e.g. equation (7), I suggest removing the x symbol.
Done: we have removed the x symbol in equation (7) and elsewhere.

page 17, line 15: what role does sliding due to subglacial hydrology play in these experiments? Here subglacial discharge primarily affects melting at the front, yet could also substantially affect sliding, which would be worth mentioning.
We thank the reviewer for pointing out that we did not discuss how surface runoff could affect basal sliding. As there is a current disagreement in the literature on the implications of this process on ice sheet evolution, ISMIP6 protocol does not include this process in our "standard experiments". The dataset provided to the modelers taking part in the "open experiment" could be used to address this question. We revised the manuscript to mention 1) that surface runoff could enhance basal sliding and 2) that this forcing is not part of the standard approach but could be included in the open approach should modelers wish to do so. This revised text is placed in the experimental protocol overview (section 2) instead of the paragraph corresponding to the review's comment, simply because the focus of section 5.2 is the oceanic forcing implementation.

page 18, line 21: I find the presentation of this conditional statement a bit odd. It is possible that this is due to the weird spacing, but my main thought is that it is not clear what the intent of the presentation. Possibly a table or flowchart describing the different retreat scenarios would be better?
We thank the reviewer for this comment, as indeed something happened in formatting of this conditional statement, and a similar comment applies for the ice_shelf_collapse_mask conditional statement (Section 6). We have removed the conditional statements and use sentences instead to describe the implementation.

5. Figure 9 and 10: these figures are extremely small and are a little difficult to read for that reason.

We have revised Figures 9 and 10.

6. page 20, section 6: how does the ongoing discussion of the `marine ice cliff instability' play into these choices?
As there is still a lot of research to better understand the marine ice cliff instability hypothesis, including how and why ice shelves collapse, our experimental protocol focusses on hydrofracturing for ice shelf collapse instead. Our choice is motivated because it is a mechanism

for ice shelf collapse that has been observed, studied and implemented in ice sheet models. This allows for an assessment of the impact of this type of ice shelf collapse on a large variety of ice flow models. This assessment becomes even more important in the ongoing discussion of marine ice cliff instability as hydrofracture is thought to be a precursor to marine ice cliff instability. However, as we do acknowledge that hydrofracturing is only one possible mechanism that can explain ice shelf collapse, we modified the text to emphasize the motivation for this experiment, as well as how our experiments fits in the ongoing discussion of marine ice cliff instability.

---

## Author Comment (AC2) · 8 May 2020

**General comments**

This paper documents the experimental setup for standalone ice sheet model intercomparison experiments using a variety of climate model forcings from CMIP6. This is a tremendous undertaking and these kinds of experiments are vital in assessing the sources of uncertainty in projections of sea-level rise over the coming century. While I recommend the paper for publication with a few revisions, I have some concerns about the design of the experiments and what they aim to test. At the same time, I realize that this project has to meet the conflicting demands of running a thorough experiment that tests a wide range of parameters, and creating a protocol that ice sheet modelers can follow.

We thank the reviewer for this review and comments, as well as for appreciation of the challenges faced by ISMIP6 in its protocol design.

**Specific comments**

The experiment aims to test the sensitivity of sea-level projects to many variables: which ice sheet model is used, which future anthropogenic forcing scenario (RCP2.6 vs 8.5) is used, which atmosphere-ocean general circulation model is used, and how the outputs of each GCM are downscaled from their native resolution to the relatively finer resolution of ice sheet models. The authors themselves state that the AOGCM forcing needs to be supplied to ice sheet models in a "uniform, standardized manner" (pg. 2 lines 29-30). While this protocol will serve as a valuable guide for future sea-level projection experiments, I think there are two respects in which the uniformity could be improved.

The experimental design uses six AOGCMs for Greenland and six for Antarctica, but only two models are common to each (CSIRO-Mk3.6 and HadGEM2-ES). Several others are clearly related, for example, MIROC5 and MIROC-ESM-CHEM, but there's nothing in the text explaining the differences. The authors cite Barthel et al. which is currently in review but some summary of the differences would be worthwhile. It's not my place to review the Barthel et al. paper, but including GCMs that meet the criteria for inclusion for only one ice sheet but not the other is a departure from the authors' stated goal of uniformity. An argument could be made here that less is more. There's a similar problem with the climate forcing scenarios – RCP2.6 with some models but not others, no intermediate climate forcing.

We thank the reviewer for this comment The model selection criteria (Barthel et al., 2020) are the same for the two ice sheets: i) present-day polar climate in agreement with observations (evaluated by model biases over the historical period, for example Agosta et al. (2015)), ii) sampling a diversity of future climate (evaluated by difference in projections and code similarities), and iii) a focus on models with RCP8.5 and RCP2.6 which also have the fields required for RCM downscaling. But is it correct that the model selection process is independent for the two ice sheets, in part because the outcome of i) is different for the two ice sheets (one climate model maybe great for Greenland but struggle for Antarctica for example, so it would make no sense to use a model that is bad for Antarctica for the sake of consistency). Barthel et al. is now published and the model selection criteria are presented in section 2. We revised the manuscript in section 2 to include a Table listing the models used in ISMIP6 (and references), as well as where to find detailed comparison of the CMIP5 models. While preparing the new Table, we noticed and rectified 2 minor mistakes in the CMIP5 acronyms, such that from the 6 models used for Greenland and 6 models used for Antarctic, 4 models are common (while our previous text implied that only 2 CMIP5 models were the same, with the 2 mistakes being IPSL-CM5-MR instead of IPSL-CM5A-MR and NorESM1 instead of NorESM1-M). The new text highlights that the selection resulted in 4 common models, one model from the same family (MIROC5 and MIROC-ESM-CHEM) and the sixth CMIP5 model being distinct for the 2 ice sheets.

The decision for a focus on RCP8.5/SSP5-8.5 is because it is anticipated that this would be the future scenario that will result in the largest sea-level contributions from ice sheets, and thus of more relevance to society in terms of planning for future sea-level rise. However, ISMIP6 decided to investigate a lower emission scenario (RCP2.6/SSP1-26) with a few CMIP models in order to capture a lower end sea level projection. Due to the time required for preparing the datasets and running the simulations (both human and computational resources), ISMIP6 felt that it could not ask more from its members given that ISMIP6 does not have any funding for participants. ISMIP6 thought that it was therefore better to sample a range of CMIP models for a given scenario in order to understand uncertainty due to climate model, and to sample the uncertainty in the parameterization instead of more RCPs.

Finally, ISMIP6 was limited by external timescale: on the one hand the CMIP6 models kept on being delayed and only became available in Summer 2019, while the IPCC deadline for paper submission remained fixed to Dec 31 2019. This means that in Fall 2018, ISMIP6 made a decision to change our protocol from the one described in Nowicki et al. (2016), and use CMIP5 models for forcing instead. The implication is that the available time for ice sheet model simulation and dataset preparation was significantly reduced.

The biggest issue I have is with the climate model downsampling. The Greenland runs use the regional climate model MAR, while Appendix C seems to say that the Antarctic climate model output was directly interpolated onto the ice sheet model grid. The authors state that using a RCM for Antarctica was prohibitively expensive. I certainly won't argue that point but several of the coauthors of this paper have run MAR for Antarctica (Agosta et al. 2019, Estimation of the Antarctic surface mass balance using the regional climate model MAR) and doing so for this study would be a big improvement. In principle the authors could test whether the downscaling or the choice of climate model had more of an effect by also interpolating the GCM output directly for Greenland and comparing the results. But this might not be very informative for Antarctica as the two continents have different topographic relief. The experiment is consistent in using the same parameterizations to extrapolate the oceanic variables for both ice sheets, and it would be great to see the same methodology applied to the atmosphere too.

We appreciate the suggestions made by the reviewer on how to improve this aspect of our protocol. Our original intention was to indeed use RCMs to downscale Antarctica SMB, but the time commitment and computational resources faced by the project in order to produce a forcing

dataset in time for ice modeling groups to run the simulations prior to the IPCC paper submission deadline made this option not possible.

Now that ISMIP6 is no longer constrained by the IPCC deadlines, we plan to offer our participants the opportunity to do the two suites of experiments that you suggest:

- 1) The use of RCM downscaled SMB for Antarctica. Our RCM team has already started preparing these datasets for selected models.
- 2) The use of Greenland SMB obtained directly from the GCM, for a few CMIP6 models that have appropriate fields for computing SMB, so that these experiments can be compared to the experiment where SMB has been downscaled with an RCM. This is an option that we had considered with CMIP5 models, but our evaluation of the CMIP5 GCMs did not result in suitable candidate: the CMIP5 models simply did not have the variables needed to compute SMB and if they did, the resulting SMB did not capture the expected large SMB gradients at the edge of the ice sheet. Some CMIP6 models show promise for this experiment due to their improved SMB and will be our focus.

These experiments will not be part of the original ISMIP6 protocol but part of ISMIP6 follow-on activities. No changes were made to the manuscript in response to this comment.

**Technical corrections**

Page 5, line 5: criteria Done

Page 6, line 6: scalar Done

Page 7, line 2: Time-dependent data assimilation does a much better job about transients, see e.g. Goldberg et al. 2015 or Gillet-Chaulet 2019.

We agree with the reviewer that time-dependent data assimilation does a better job about transients, and this is the way that the community will likely initialize ice sheet models in the future. We note that Goldberg et al. 2015 is a regional study and Gillet-Chaulet 2020 is a synthetic flow line set-up, such that time-dependent data assimilation on ice sheet wide scale is not yet possible due to the lack of ice sheet wide observations and computational challenges. Nonetheless, we have added the following sentence: "Time-dependent data assimilation methods allow for more realistic transients, but to date have been limited to regional studies or synthetic ice sheet setup (e.g. Goldberg et al., 2015; Gillet-Chaulet, 2020)"

Note: we are assuming that the reviewer is referring to this paper: Gillet-Chaulet, F.: Assimilation of surface observations in a transient marine ice sheet model using an ensemble Kalman filter, The Cryosphere, 14, 811–832, https://doi.org/10.5194/tc-14-811-2020, 2020. Figures 2-5: These figures are difficult to parse visually. Some way of showing the difference between the RCP2.6 and RCP8.5 scenarios would be especially helpful, either by using dashed lines for one scenario in the same plot or, better yet, putting the two on different plots entirely. We have redone these figures to highlight the differences between the RCP2.6 and RCP8.5 scenarios by using dashed lines for RCP2.6.

Page 18: Several paragraphs repeat information that's already in Slater et al. This paper is long as it is and a shorter summary of this would cut down on length some. Likewise the discussion of the results from Jourdain et al.

We thank the reviewer for this comment. While we recognize that the manuscript is long, this is a necessity in order to highlight the connections of all the components of the ISMIP6 protocol, as well as having one manuscript describing in details the ISMIP6 protocol (a requirement for MIPs endorsed by CMIP6). We have paid attention to keep the descriptions to a minimum and highlight in the figures aspects not shown in other ISMIP6 publications. The only panel that is in common with Slater et al. and Jourdain et al. is the timeseries panels Fig 7a, 8a, 9a, which have been augmented with CMIP6 models, as these were not shown in Slater et al. and Jourdain et al. Reviewer 1 has asked us the opposite: to explain the results of all ISMIP6 cited papers to a greater degree (see 3rd remark).

Page 21, line 8: the second period... lasts Done